# EgoDTM: Towards 3D-Aware Egocentric Video-Language Pretraining

**Boshen Xu**[1]  **Yuting Mei**[1]  **Xinbi Liu**[1]  **Sipeng Zheng**[2]  **Qin Jin**[1*]
[1] AIM3 Lab, Renmin University of China    [2] BeingBeyond

## Abstract

Egocentric video-language pretraining has significantly advanced video representation learning. Humans perceive and interact with a fully 3D world, developing spatial awareness that extends beyond text-based understanding. However, most previous works learn from 1D text or 2D visual cues, such as bounding boxes, which inherently lack 3D understanding. To bridge this gap, we introduce **EgoDTM**, an **Ego**centric **D**epth- and **T**ext-aware **M**odel, jointly trained through large-scale 3D-aware video pretraining and video-text contrastive learning. EgoDTM incorporates a lightweight 3D-aware decoder to efficiently learn 3D-awareness from pseudo depth maps generated by depth estimation models. To further facilitate 3D-aware video pretraining, we enrich the original brief captions with hand-object visual cues by organically combining several foundation models. Extensive experiments demonstrate EgoDTM's superior performance across diverse downstream tasks, highlighting its superior 3D-aware visual understanding. Code: https://github.com/xuboshen/EgoDTM.

## 1 Introduction

The development of embodied AI capable of fulfilling diverse societal roles has long been a goal in artificial intelligence [25, 49]. A promising path to achieving this vision involves developing egocentric AI, which can comprehend human activities by analyzing large-scale egocentric videos captured from a first-person perspective using wearable devices. These videos provide rich insights into hand-object interactions (HOI) [19, 79, 11, 39], offering a window into how individuals interact with nearby objects through their hands. With the emergence of large-scale egocentric datasets [22], video-language pretraining [36, 87] has become a dominant paradigm for learning egocentric video representations, significantly improving performance on downstream tasks such as video-text retrieval [11, 67] and action recognition [66, 34].

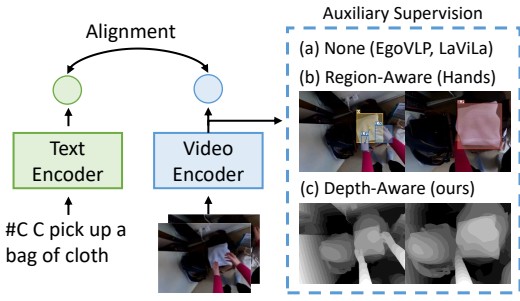

Figure 1: Comparison of egocentric pretraining paradigms. While previous paradigms focus on text-based [36, 51, 87] or 2D spatial region-aware learning [80], EgoDTM incorporates 3D spatial information to enhance video representations.

Humans possess an innate ability to perceive and reason about 3D spatial relationships, effortlessly perceiving relative distances and spatial arrangements from visual cues alone [68, 28, 65]. While there is no universal definition of a model's 3D awareness, we define it as the ability to infer 3D

---

*Qin Jin is the corresponding author.

39th Conference on Neural Information Processing Systems (NeurIPS 2025).

information from 2D images. However, models pretrained solely on 2D frames and general-purpose text (e.g., CLIP [52]) often struggle to develop robust 3D awareness [18, 42, 56]. To equip egocentric models with such 3D awareness, we wonder: Can video-language models better understand egocentric contexts by incorporating 3D-aware perception?

Achieving this goal presents two key challenges. First, 3D representations are diverse, and obtaining corresponding labels is often costly. Recent works [78, 84] explore using multi-view images for 3D reconstruction to distill 3D awareness into 2D models. However, in egocentric scenarios, depth maps are more practical and easier to obtain. Depth maps provide direct 3D distance information and help distinguish salient objects from the background, offering crucial cues for spatial understanding. Unfortunately, existing egocentric datasets with depth maps [39, 55, 29] remain limited in both scale and diversity for large-scale pretraining. Recent foundation models for depth estimation [76, 77, 5] demonstrate strong out-of-domain generalization, enabling us to construct a large-scale depth-augmented egocentric dataset at a low cost.

Second, effective 3D-aware pretraining requires bridging the modality gap between depth and text. As illustrated in Figure 1, common video-language pretraining [36, 51, 87] primarily relies on textual supervision, thus avoiding this challenge. Recent works [80, 47] explore region-aware video-language pretraining using non-pixel-level cues, such as text and sparse object bounding boxes, which face only minor modality gaps. However, unlike these non-pixel-level cues, regressing the pixel-level outputs (e.g., depth maps) requires fundamentally different capabilities compared to predicting texts or bounding boxes, as suggested by previous studies [30, 31]. Additionally, depth estimation typically requires high-resolution inputs and multi-scale features, whereas video-language models rely on large batch sizes for contrastive learning, making direct integration nontrivial. Thus, designing an effective learning approach and enriching textual supervision with spatial information are crucial for successful 3D-aware pretraining.

To address these aforementioned challenges, we introduce **EgoDTM**, a novel 3D-aware egocentric video-language model that learns video representation from depth maps and spatially informed captions. In addition to dual transformer encoders for video-text alignment, EgoDTM adopts a 3D-aware module for video pretraining and a data construction pipeline to enrich captions with spatial information. To adapt the 3D-aware module for the video-language framework, we propose a unified visual representation with a lightweight depth decoder, supervised by depth predictions from foundation models [76]. Specifically, the lightweight depth decoder takes as input the visual representation from the last layer of the video encoder and predicts the composition of discrete adaptive bins [3, 4, 64, 35] to estimate low-resolution depth maps. Moreover, we adopt off-the-shelf visual foundation models [63, 59] to create high-quality HOI masks by first detecting HOIs, then selecting key frames, and finally tracking bidirectionally. To enrich the original captions with spatial information, we leverage a large language model (LLM) guided by the temporally consistent HOI masks. Through 3D-aware video-language pretraining, EgoDTM improves visual generalization on downstream tasks involving egocentric HOIs.

Our contributions are threefold: (1) We introduce EgoDTM, a 3D-aware egocentric video-language model learned from 3D-aware video-language pretraining. (2) We develop a lightweight 3D-aware decoder for depth estimation and a data construction pipeline to enrich captions with spatial information. As a byproduct, we generate millions of egocentric data, including captions, HOI boxes, HOI masks, and depth maps. (3) Extensive experiments demonstrate that EgoDTM significantly enhances performance on video understanding tasks like video-text matching, and 3D understanding tasks like robot manipulation.

## 2 Related Works

**Egocentric Video-Language Pretraining.** Egocentric video understanding [22, 8, 50, 83, 43, 74, 11, 73, 75, 33, 72] typically involves human daily interaction between hands and objects from a first-person perspective. Inspired by visual-language pretraining paradigms in third-person datasets [2, 41, 52, 47, 80], EgoVLP [36] firstly proposes to conduct egocentric video-language pretraining on the large-scale egocentric dataset Ego4D [22], which aims to learn video representations from massive video-text data via contrastive learning. Since EgoVLP, similar paradigms [36, 51, 87, 1] have gained large success in EgoHOI understanding, For example, LaViLa [87] employs a video-conditioned GPT-2 [53] and a T5 [54] rephraser to generate text descrip-

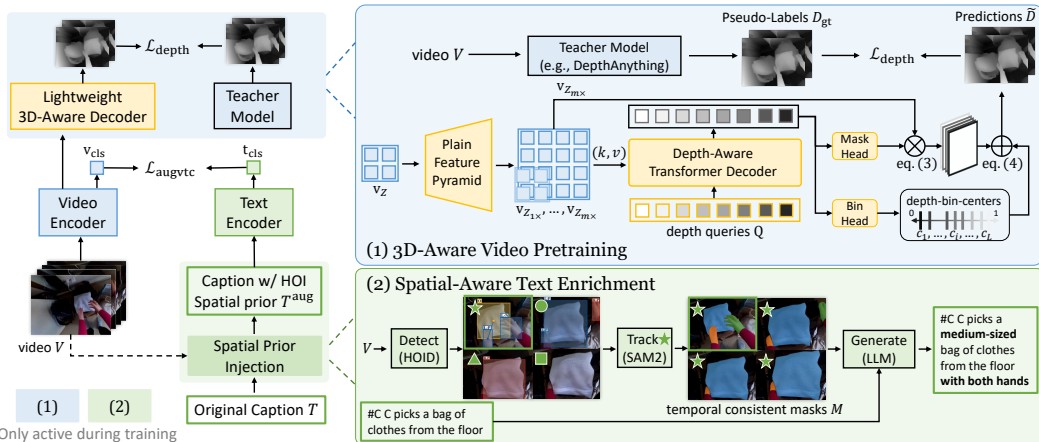

Figure 2: EgoDTM learns 3D-aware representations from depth and text. Our dual encoders are constructed using only transformers [16, 70, 52] with flash attention [12]. During pretraining, we conduct (1) 3D-aware video pretraining: we design a lightweight 3D-aware decoder to predict depth using visual feature maps, supervised by a teacher foundation model [77]. The decoder contains a plain feature pyramid to get multi-scale features, a depth-aware transformer decoder to process depth queries with video features, and the heads to predict depth maps; (2) Spatial-aware textual enrichment: we enhance captions with spatial information by organically combining foundation models in the detect-track-generate pipeline. Different green markers denote inconsistency of HOI predictions; identical ones indicate consistency.

tions for egocentric videos to expand video-text data. Despite the progress, pretraining with text alone often lacks precision in target localization. In response, some studies have focused on developing region-aware representations for EgoVLMs. For example, HelpingHands [80] proposes learning from noisy hand-object detection results, while HENASY [47] ensembles an additional hierarchical encoder to learn HOI region-aware representation. However, these methods are limited to 2D reasoning and lack an understanding of real-world 3D context. In this work, we take a step towards 3D-aware egocentric video-language pretraining.

**3D-Awareness of 2D Vision Models.** As suggested by research in developmental psychology and psychophysics, we humans are capable of understanding 3D information, like depth and orientations, from only 2D visual signals [68, 28, 65]. With the growing interest in embodied intelligence [25, 44] and personal AI assistants [49], the demand for 3D-aware 2D visual models has become increasingly important. Although there is no widely accepted consensus to define and build effective 3D-aware models, some studies have started to assess the 3D awareness of visual foundation models. For example, Probing3D [18] demonstrated that CLIP, which is pretrained exclusively with text, is significantly limited in predicting depth and 3D surface normals. EysWideShut[69] revealed that CLIP struggles with 3D-related tasks, such as recognizing object orientation and understanding compositional contexts. More recently, relevant works have made endeavors to enable visual foundation models (e.g., CLIP) with 3D awareness by fine-tuning on 3D-aware data. For instance, FiT3D [78] finetunes visual foundation models with 3D Gaussian features reconstructed from multi-view images. SpatialVLM [9] conducts visual instruction tuning on metric depth-aware QA pairs for multi-modal large language models. Different from these learning paradigms, we define 3D-awareness as the model's latent ability to estimate depth and pretrain an egocentric video-language model with a lightweight 3D-aware decoder.

## 3   Method

Our objective is to develop a 3D-aware egocentric video-language model, where 3D-awareness is defined as the model's ability to infer depth information from its representations. EgoDTM achieves this by pretraining depth-aware and text-aligned video representations.   Figure 2 illustrates the architecture of our proposed EgoDTM. We first describe the video-language model architecture in Section 3.1. Next, we introduce the 3D-aware video pretraining in Section 3.2 and then present

our approach to generate spatial-aware captions in Section 3.3. The details of training and inference are presented in Section 3.4.

## 3.1 Video-Language Model Architecture

The video-language model basically consists of a video encoder and a text encoder.

**Video and text encoders.** Our video encoder employs a plain vision transformer [16] backbone to process video tokens. The video input $V \in \mathbb{R}^{F \times H \times W \times 3}$ is divided into $N = \frac{F}{F_0} \times \frac{H}{H_0} \times \frac{W}{W_0}$ non-overlapping cubes of dimension $F_0 \times H_0 \times W_0 \times 3$, where $F$ is the number of frames, $H$ and $W$ denote height and width. These cubes are combined with positional embeddings and processed by the video encoder to produce a feature map $\mathbf{v}_Z \in \mathbb{R}^{N \times C}$. Note that a [CLS] token is added to represent the global video embedding $\mathbf{v}_{\mathrm{cls}}$. For the text encoder, captions $T$ are tokenized via a Byte Pair Encoding (BPE) tokenizer [62] and encoded with a transformer initialized from CLIP [52] to output the sentence embedding $\mathbf{t}_{\mathrm{cls}}$. Both encoders adopt the flash-attention [12] to mitigate the memory bottleneck of the attention mechanism.

**Video-text alignment loss.** We use the InfoNCE loss to align video and text embeddings. For simplicity, we define the video-to-text loss within a batch $\mathcal{B} = \{(\mathbf{v}_i, \mathbf{t}_i\}_{i=1}^B$ as follows:

$$\mathcal{L}_{\mathrm{vtc}} = -\frac{1}{B} \sum_i \log \frac{\exp(\mathbf{v}_i \cdot \mathbf{t}_i / \tau)}{\Sigma_j \exp(\mathbf{v}_i \cdot \mathbf{t}_j / \tau)} \tag{1}$$

where $(\mathbf{v}_i, \mathbf{t}_i)$ represent the video and text embeddings $(\mathbf{v}_{\mathrm{cls}}, \mathbf{t}_{\mathrm{cls}})$ for the $i$-th (video, caption) pair within the batch. The text-to-video loss is defined symmetrically.

## 3.2 3D-Aware Video Pretraining

A critical issue in leveraging depth maps for 3D-aware pretraining is that depth maps are typically high-resolution images (e.g., 1024p). However, most low-level details in these maps are redundant, as video-language models primarily benefit from the relative depth information between patches. Therefore, high-resolution depth maps may be unnecessary and inefficient for pretraining. Additionally, video-text pretraining often requires a large batch size to ensure good performance, further emphasizing the need for lightweight and memory-efficient processing. To address these concerns, we design a lightweight 3D-aware decoder to estimate low-resolution depth maps from video representations. Specifically, the decoder consists of a plain feature pyramid, a depth-aware transformer decoder, and two specialized heads, as described below.

**Plain feature pyramid.** Depth estimation methods [48, 3, 57] often utilize multi-scale features from intermediate backbone layers. However, video-language pretraining typically uses only the final-layer representations. To reconcile these differences, we adopt a simplified FPN [32] that projects the video feature map $\mathbf{v}_Z$ into multi-scale outputs $\{\mathbf{v}_{Z_{1\times}}, \mathbf{v}_{Z_{2\times}}, \cdots, \mathbf{v}_{Z_{m\times}}\}$, where $\mathbf{v}_{Z_{m\times}}$ denotes the feature map of size $(N \times m^2) \times C$, and the size of the predicted depth maps equals $N \times m^2$. Moreover, inspired by the adaptive bins [3, 4, 64, 35] in depth estimation, we segment the depth range into intervals and predict distributions at each pixel over the interval centers.

**Depth-aware transformer decoder.** This module processes multi-scale features to generate depth-aware representations capable of estimating distinct distance ranges. The transformer decoder follows standard architecture as [7, 10], transforming $L$ learnable depth queries $\mathbf{Q} \in \mathbb{R}^{L \times C}$ into depth-aware representations $\mathbf{Z} \in \mathbb{R}^{L \times C}$ using self-attention and cross-attention mechanisms. The cross attention processes $\mathbf{Q}$ as queries and concatenates $(\mathbf{v}_{Z_{1\times}}, \mathbf{v}_{Z_{2\times}}, \ldots, \mathbf{v}_{Z_{m-1\times}})$ as key-value pairs $(k, v)$. The learned depth queries estimate depths at different distances, as illustrated by varied gray colors in Figure 2.

**Bin head.** We partition the depth range of $[0, 1]$ into $L$ intervals. The bin head $\mathcal{F}(\cdot) : \mathbb{R}^C \to [0, 1]$ processes depth-aware representations $\{\mathbf{z}_i\}_{i=1}^L$ to predict widths of depth interval bins $\{w_i\}_{i=0}^L$, satisfying $\sum_{i=0}^L w_i = 1$ where we denote $w_0 = 0$. The $i$-th depth interval is defined by boundaries $[\sum_{i=1}^{i-1} w_i, \sum_{i=1}^{i} w_i]$, with its center computed as:

$$c_i = \sum_{i=1}^{i-1} w_i + \frac{w_i}{2} \tag{2}$$

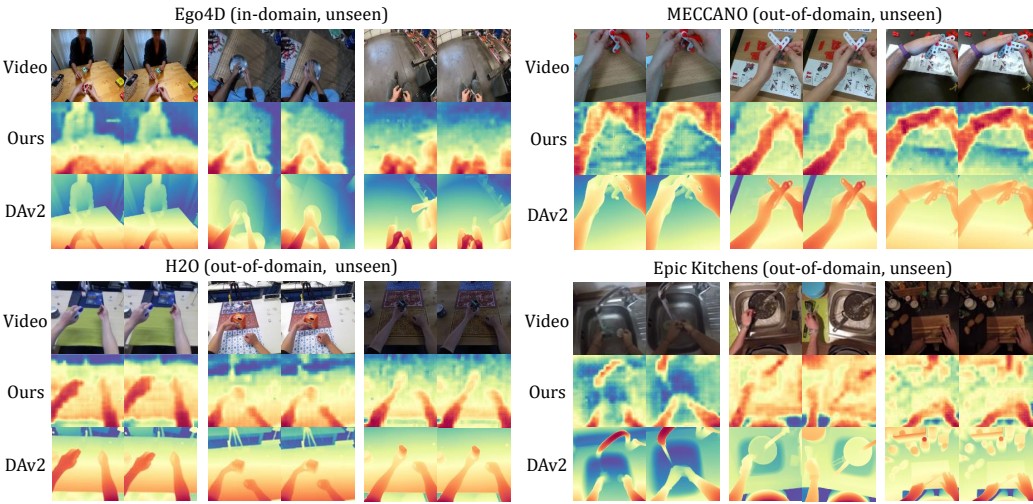

Figure 3: Qualitative results of depth estimation from EgoDTM and DepthAnythingv2 [77](DAv2) on datasets including in-domain but unseen Ego4D validation set [22], out-of-domain and unseen data of EK100 [11], MECCANO [55], and H2O [29]. Note that DAv2 operates with a high resolution of 512p, while EgoDTM uses a lower resolution input of 224p and generates a depth map at a resolution of 56p. Despite the lower resolution input, EgoDTM demonstrates intuitive generalization across unseen egocentric datasets with diverse environments, illuminations, backgrounds, and varying HOI object sizes.

**Mask head.** Then, we aim to predict distributions over the depth centers at each pixel. Specifically, the mask head $\mathcal{G}(\cdot) : \mathbb{R}^C \to \mathbb{R}^C$ takes $\mathbf{z}_i$ as input, and the output $\mathbf{m}_i$ is multiplied by the largest feature map $\mathbf{m}_i \otimes \mathbf{v}_{Z_{m\times}}$ to acquire unnormalized probabilistic maps $d_i \in \mathbb{R}^{N \times m^2}$, where $\otimes$ denotes the composition of element-wise multiplication and sum along the feature dimension. After normalizing $\{d_i\}_{i=1}^L$ by the Softmax operation along the query dimension $L$, we get the probabilistic map $D = [D_1, \ldots, D_L] \in \mathbb{R}^{L \times N \times m^2}$. The process is described as:

$$D = \mathrm{Softmax}(\mathcal{G}(\mathbf{Z}) \otimes \mathbf{v}_{z_{m\times}}) \tag{3}$$

Finally, depth prediction $\tilde{D} \in \mathbb{R}^{N \times m^2}$ is obtained by the linear combination of the depth-bin-centers and the probabilistic maps:

$$\tilde{D} = \sum_{k=1}^{L} c_k D_k \tag{4}$$

**3D-aware pretraining loss.** Given the lack of depth annotations in large-scale egocentric videos [22], we generate pseudo-depth labels using monocular depth estimation foundation models [57, 76, 77]. The pretraining loss is defined as:

$$\mathcal{L}_{\mathrm{depth}} = \|\tilde{D} - D_{\mathrm{gt}}\|_2 \tag{5}$$

where $\tilde{D}$ and $D_{\mathrm{gt}}$ are the predicted and ground-truths inverse depth maps within the range $[0, 1]$.

### 3.3 Spatial-Aware Textual Enrichment

Previous egocentric video-language pretraining frameworks typically learn from short descriptions composed of verbs and nouns. To foster 3D-aware pretraining, we argue that captions should contain more spatial-temporal information like HOI positions, object shapes, and movements. We therefore enrich the captions with visual cues by employing a detect-track-generate pipeline tailored for egocentric videos using foundation models [63, 59, 14].

**Detect.** We use the detector [63], HOID, finetuned on egocentric data to detect hands, objects, and their contact states for a given frame. The detector $f(\cdot)$ receives $V_i$ as input and outputs diverse

| | Epic-Kitchens-100-MIR | | | | | | EGTEA | | EgoMCQ | |
|---|---|---|---|---|---|---|---|---|---|---|
| Method | mAP (%) | | | nDCG (%) | | | mean | top1 | Inter | Intra |
| | V→T | T→V | Avg. | V→T | T→V | Avg. | | | | |
| EgoVLP [36] | 26.0 | 20.6 | 23.3 | 28.8 | 27.0 | 27.9 | - | - | 90.6 | 57.2 |
| EgoVLPv2 [51] | 35.1 | 26.6 | 30.8 | 33.7 | 30.4 | 32.0 | 30.9 | 35.1 | 91.0 | 60.9 |
| LaViLa [87] | 35.1 | 26.6 | 30.8 | 33.7 | 30.4 | 32.0 | 30.9 | 35.1 | 93.6 | 59.1 |
| AVION [86] | 37.1 | 28.7 | 32.9 | 34.4 | 31.0 | 32.7 | 38.6 | 42.3 | 94.4 | 62.1 |
| HelpingHands* [80] | 35.6 | 26.8 | 31.2 | 34.7 | 31.7 | 33.2 | 29.4 | 35.3 | 93.2 | 58.8 |
| HENASY* [47] | 35.5 | 27.1 | 31.3 | 34.6 | 31.7 | 33.2 | 29.6 | 35.9 | 94.1 | 61.3 |
| EgoDTM (ours) | **37.9** | **29.1** | **33.5** | **34.8** | **31.9** | **33.4** | **40.2** | **43.2** | **94.6** | **63.6** |

Table 1: Comparison with state-of-the-art methods on zero-shot video-text retrieval and action recognition. All methods use the same 4M videos from Ego4D for pretraining and have similar hidden dimension sizes. The numbers of the method with * are sourced from [47].

information $\{(b_j, c_j, t, s)\}_j$, where $b_j$ is the bounding boxes with confidence scores $c_j$, $t$ is object type, and $s$ is contact state of hand.

**Track.** Since predictions of HOID lack temporal consistency, we address this issue by tracking across the video. We utilize SAM2 [59] $g(\cdot)$, a foundation model for promptable segmentation and tracking. The initial tracking frame $V^*$ is selected when the hands make contact with objects. The HOI masks of this frame are obtained by prompting SAM2 with HOI boxes $\{m_j^*\}_j = g(V^*, \{b_j^*\}_j)$, where the bounding boxes serve as negative prompts for each other. Finally, the model tracks HOI masks throughout all frames to acquire the temporal consistent masks $M = (\{m_{1j}\}_j, \dots, \{m_{Fj}\}_j)$.

**Generate.** As suggested by previous works [37, 20], LLMs are capable of understanding visual cues like bounding boxes. We use an LLM $h(\cdot)$ to comprehend the HOI spatial information and original captions, then generate spatial-aware captions $T^{aug} = h(T, M)$.

Equipped with the enriched video-text pairs, our new video-text contrastive learning objective is:

$$\mathcal{L}_{augvtc} = -\frac{1}{B}\sum_i \log\frac{\exp(\mathbf{v}_i \cdot \mathbf{t}'_i)/\tau)}{\Sigma_j\exp(\mathbf{v}_i \cdot \mathbf{t}'_j/\tau)} \tag{6}$$

where $\mathbf{t}'_i = \text{rand}(\mathbf{t}_i, \mathbf{t}_i^{aug})$ is randomly sampled from the original and enriched text embedding $\mathbf{t}_i^{aug}$.

### 3.4 Training Strategy

**Training and evaluation.** We employ both the augmented video-text alignment loss and the 3D-aware pretraining loss to pretrain our model:

$$\mathcal{L} = \mathcal{L}_{augvtc} + \mathcal{L}_{depth} \tag{7}$$

Our pretraining improves video-language representations for downstream egocentric tasks without modifying the dual-encoders themselves. As a result, only the dual-encoders are used for downstream tasks, introducing no additional computational costs at inference time.

**Pretraining data.** Our pretraining data consists of four million (video, text) pairs, with each video approximately 1 second long. We generate enhanced text descriptions and depth maps specifically for videos featuring hand-object interactions, resulting in two million (video, enriched texts, depths) triplets. Specifically, for each video, we extract eight depth maps using the DAv2-Large [77] model. The HOI boxes are detected by HOID [63], which is finetuned on egocentric data. The HOI masks are segmented and tracked using the same unified SAM2-Large model [59]. The final enriched text descriptions are generated using the DeepSeek-LLM-200B model [14].

## 4 Experiments

To evaluate our EgoDTM, we conduct experiments from three perspectives: short video understanding (video-text retrieval and action recognition), 3D space comprehension (depth estimation and robot manipulation), and long video understanding (natural language query and moment query). These evaluations cover seven benchmarks across five datasets. In the following sections, we detail

| Method | Scale-Aware Metrics | | | | Scale-Invariant Metrics | | | |
|---|---|---|---|---|---|---|---|---|
| | $\delta_1\uparrow$ | $\delta_2\uparrow$ | $\delta_3\uparrow$ | RMSE$\downarrow$ | $\delta_1\uparrow$ | $\delta_2\uparrow$ | $\delta_3\uparrow$ | RMSE$\downarrow$ |
| ConvNext [40] | 0.721 | 0.965 | 0.991 | 0.644 | 0.727 | 0.969 | 0.996 | 0.593 |
| CLIP [52] | 0.795 | **0.966** | 0.988 | 0.624 | 0.811 | 0.976 | 0.994 | 0.559 |
| EgoVLP [36] | 0.778 | 0.954 | 0.989 | 0.610 | **0.853** | **0.977** | 0.996 | 0.497 |
| LaViLa [87] | 0.801 | 0.954 | 0.987 | 0.598 | 0.811 | 0.964 | 0.993 | 0.552 |
| AVION [86] | 0.786 | 0.960 | 0.991 | 0.606 | 0.812 | 0.969 | 0.996 | 0.543 |
| EgoDTM (ours) | **0.826** | 0.964 | **0.993** | **0.539** | 0.848 | **0.977** | **0.998** | **0.481** |

Table 2: Comparisons of depth estimation task on H2O dataset.

the experimental setups (Section 4.1), main results (Section 4.2), ablation studies (Section 4.3), and further analyses (Section 4.4).

## 4.1 Benchmarks and Settings

**Zero-shot video-text retrieval (ZS-VTR).** To assess video-text alignment, we conduct zero-shot retrieval evaluations on text-to-video multi-choice retrieval on Ego4D [22] (EgoMCQ) and multi-instance video-text retrieval on Epic-Kitchens-100 [11] (EK100MIR). Following [36, 11], we use mean Average Precision (mAP) and the normalized Discounted Cumulative Gain (nDCG) as metrics for EK100MIR and accuracy for EgoMCQ.

**Zero-shot action recognition (ZS-AR).** Action recognition is conducted in a video-to-text retrieval manner. We experiment on the EGTEA [34] and Epic-Kitchens-100 [11] (EK100CLS). The task on EGTEA requires models to recognize 106 classes of cooking activities, and the EK100CLS includes evaluation on 97 verbs and 300 nouns in kitchens. The metrics include mean accuracy, top-1 and top-5 accuracy across test splits.

**Depth estimation (DE).** To assess the model's 3D awareness, we conduct depth estimation by fine-tuning our model on the H2O [29] dataset. Following [18], we add a prediction head composed of a linear layer and an upsampling convolution layer on top of the frozen video encoder. Similar to [18, 17], the metrics are threshold accuracy ($\delta_i$): percentage of pixels $\tilde{D}_j$ in ground-truth depth maps that satisfy $\max(\frac{D_{\mathrm{gt}j}}{\tilde{D}_j}, \frac{\tilde{D}_j}{D_{\mathrm{gt}\ j}}) = \delta_i < thr^i$ for $thr = 1.25, i \in \{1, 2, 3\}$, and RMSE: $\|\tilde{D} - D_{\mathrm{gt}}\|_2$. The scale-invariant metrics are shifted and normalized from scale-aware metrics.

**Robot Manipulation (RM).** The robot must learn to accurately perceive and interact in 3D spaces to accomplish daily tasks. Following previous works [45, 26], we evaluate the visual representations as frozen perception modules for downstream policy learning within the Franka Kitchens simulation environment [23]. Five tasks are adopted: turn knob (TK), open door (OD), flip switch (FS), open microwave (OM), and slide door (SD). We use success rate as the metric.

**Natural language queries (NLQ).** The NLQ task localizes time intervals in a long video given a language query. We experiment on the EgoNLQ task on Ego4D [22]. To fairly evaluate different pretrained visual representations, we extract the visual features by video encoders and text features by the same pretrained BERT [15], then train a VSLNet [81] to solve this task. The evaluation metrics are "Rn@m", where $n \in \{1, 5\}$ and $m \in \{0.3, 0.5\}$, presenting the percentage of at least one of the top-$n$ predicted intervals having IoU greater than $m$.

**Moment Queries (MQ).** The MQ task is a video-only problem that aims to detect all temporal activity intervals in a long video given a specified activity category. We conduct experiments on the EgoMQ benchmark from Ego4D [22]. Following the setup in [36], we extract visual features using video encoders and then train a VSGN model [85] to perform this task. The evaluation metrics are "R@n, m", consistent with those used in the NLQ task.

**Pretraining details.** The dual-encoders are initialized by the checkpoint pretrained on the original four million video-text pairs [36] from Ego4D. EgoDTM is then trained for two epochs on 8*A800 GPUs, which requires approximately 10 hours and a learning rate of 3e-5. The depth-aware transformer decoder comprises six layers. The hidden dimension of the dual encoders is 768, while the 3D-aware decoder uses a dimension of 256 for efficient design. We use frames with 224p as input

| Method | R1@0.3 | R5@0.3 | R1@0.5 | R5@0.5 |
|---|---|---|---|---|
| EgoVLP [36] | 6.32 | 13.84 | 3.41 | 8.80 |
| LaViLa [87] | 7.12 | 14.82 | 3.87 | 9.55 |
| AVION [86] | 7.33 | 14.89 | 4.31 | 9.53 |
| EgoDTM (ours) | **8.13** | **16.11** | **4.83** | **10.30** |

Table 3: Comparisons of NLQ task.

| Method | R1@0.3 | R5@0.3 | R1@0.5 | R5@0.5 |
|---|---|---|---|---|
| EgoVLP [36] | 30.44 | 46.66 | 22.41 | 35.75 |
| LaViLa [87] | 32.9 | 48.68 | **24.12** | 37.59 |
| AVION [86] | 32.17 | 47.3 | 23.11 | 36.3 |
| EgoDTM (ours) | **32.92** | **50.08** | 23.94 | **39.15** |

Table 4: Comparisons of MQ task.

and 56p as output of the depth maps. Consequently, our 3D-aware decoder only has 9M parameters, and the batch size is set to 4096.

## 4.2 Main Results

**Zero-shot video-text retrieval.** In Table 1, EgoDTM outperforms models like LaViLa [87] and AVION [86]. For example, on EK100-MIR, EgoDTM outperforms the state-of-the-art AVION by +0.9% in mAP and +0.3% in nDCG, while also achieving a +1.5% improvement in intra accuracy on EgoMCQ. This indicates that captions generated based on visual cues from our data curation pipeline are more accurate and informative than captions generated from visual-conditioned LLMs or simply text rewriting. Besides, our model surpasses HelpingHands [80] and HENASY [47], which rely on noisy HOI detection supervision with extra parameters in the visual encoder, indicating that depth modality can potentially offer larger merits for video-language models.

**Zero-shot action recognition** Table 1 illustrates the superior performance of EgoDTM on EGTEA. While previous works have demonstrated that depth modality enhances traditional action recognition tasks [82, 38], their applicability has been largely constrained to specific scenarios and closed-set datasets. Our results extend this understanding, showing that video-language models can effectively leverage depth modality to learn more generalizable video representations.

**Depth estimation.** We evaluate the ability of visual-language models to infer depth from images in Table 2. Our model surpasses previous video-language models across most metrics by a large margin. In particular, our method improves the scale-aware RMSE by approximately 9.8% over the second-best result achieved by LaViLa. Interestingly, EgoVLP performs better in scale-invariant $\delta_1$ metric, suggesting that it latently captures finer-grained pixel-level details. We hypothesize that this advantage stems from EgoVLP's scene-aware negative sampling strategy, which samples video-text pairs captured from the same environments into the same batch, thereby learning relevant details.

**Robot manipulation.** As shown in Table 5, EgoDTM consistently outperforms pretrained visual-language models such as CLIP [52] and LaViLa [87] by +20.2% and +6.6%, respectively, demonstrating stronger spatial perception in visual representations. Additionally, EgoDTM performs competitively with specialized robot learning methods on certain tasks, such as "turn knob" but underperforms on others, like "open microwave". A possible reason is that our model, pretained on depth and text, may overfit to real-worold scenarios, whereas methods like R3M [45] leverage self-supervised pretraining, which could provide better generalization to diverse manipulation tasks.

| Method | TK | OD | OM | FS | SD | Average |
|---|---|---|---|---|---|---|
| R3M [45] | 53.3% | 50.7% | 59.3% | 86.3% | 97.7% | 69.4% |
| MPI [26] | 83.3% | 54% | 44.5% | 93.5% | 100% | 75% |
| ResNet [24] | 28% | 18% | 26.7% | 50% | 75.5% | 39.7% |
| CLIP [52] | 26.3% | 13% | 24.7% | 41.7% | 86.3% | 38.4% |
| LaViLa [87] | 48% | 26% | 22.5% | 69% | **94.5%** | 52% |
| EgoDTM (ours) | **56%** | **28%** | **35.5%** | **81%** | 92.5% | **58.6%** |

Table 5: Comparison of robot manipulation tasks in Franka Kitchen simulation to assess model's 3D-awareness.

**Natural language queries.** As shown in Table 3, EgoDTM outperforms other egocentric video-language models, e.g., +1.22 on R5@0.3 over AVION. In long video localization tasks, depth awareness enhances spatial understanding, enabling EgoDTM to better comprehend visual content.

**Moment queries.** As shown in Table 4, EgoDTM achieves competitive performance among all methods. Specifically, it surpasses AVION by +2.78 on R5@0.3 and +2.85 on R5@0.5, and slightly improves over LaViLa by +1.4 on R5@0.3 and +1.56 on R5@0.5. These results demonstrate that depth-aware representation learning in EgoDTM effectively enhances spatial reasoning and temporal localization in long egocentric videos, leading to more accurate moment predictions.

|  | EK100MIR | | EgoMCQ | |
|---|---|---|---|---|
|  | mAP | nDCG | Inter | Intra |
| $\mathcal{L}_{vtc}$ | 29.7 | 30.7 | 94.2 | 60.2 |
| $\mathcal{L}_{vtc} + \mathcal{L}_{depth}$ | 31.3 | 31.2 | 94.2 | 62.6 |
| $\mathcal{L}_{augvtc}$ | 31.3 | 32.2 | 94.0 | 61.6 |
| $\mathcal{L}_{augvtc} + \mathcal{L}_{depth}$ | 33.1 | 33.1 | 94.6 | 62.6 |

(a) **EgoDTM's components.** Both the 3D-aware video pretraining and textual enhancement enhance video-language representations.

|  | EK100MIR | | EgoMCQ | |
|---|---|---|---|---|
|  | mAP | nDCG | Inter | Intra |
| 4 | 32.6 | 32.5 | 94.61 | 62.61 |
| 8 | 33.1 | 33.1 | 94.6 | 62.64 |
| 16 | 32.3 | 32.4 | 94.72 | 62.46 |

(b) **Depth query numbers.** We set 8 queries as the default, which empirically generalizes best on EK100MIR.

| Reso | Mem | EK100MIR | | EgoMCQ | |
|---|---|---|---|---|---|
|  |  | mAP | nDCG | Inter | Intra |
| 28p | 10G | 30.4 | 31.2 | 94.54 | 60.65 |
| 56p | 17G | 30.9 | 31.9 | 94.28 | 61.05 |
| 112p | 59G | 31.2 | 31.9 | 94.37 | 60.38 |

(c) **Resolution of depth.** The batch size is set to 512 to accommodate the high GPU memory demands of large depth maps.

|  | EK100MIR | | EgoMCQ | |
|---|---|---|---|---|
|  | mAP | nDCG | Inter | Intra |
| Base | 33.1 | 33.1 | 94.6 | 62.64 |
| Large | 36.4 | 34.2 | 95.13 | 66.04 |

(d) **Model size.** Large model brings more performance gain on intra accuracy of EgoMCQ.

|  | EK100MIR | | EgoMCQ | |
|---|---|---|---|---|
|  | mAP | nDCG | Inter | Intra |
| All | 32.8 | 32.7 | 94.46 | 61.92 |
| Rand | 33.1 | 33.1 | 94.6 | 62.64 |

(e) **Text Sampling Strategy.** Random substitution is better than thorough replacement.

|  | EK100MIR | | EgoMCQ | |
|---|---|---|---|---|
|  | mAP | nDCG | Inter | Intra |
| 1024 | 32.4 | 32.5 | 94.55 | 61.48 |
| 2048 | 33.1 | 33.1 | 94.6 | 62.64 |
| 4096 | 33.5 | 33.4 | 94.54 | 63.55 |

(f) **Batch Size.** Increasing the batch size steadily yields larger improvements.

Table 6: Zero-shot ablation studies. Models are pretrained on 4M video-text pairs from Ego4D and evaluated with zero-shot video-text retrieval on Epic-Kitchens-100-MIR [11] and EgoMCQ [36]. Unless specified, the default setting includes: ViT-B/16 backbone, batch size of 2048, random substitution of enriched texts, both $\mathcal{L}_{depth}$ and $\mathcal{L}_{augvtc}$ as losses, 8 depth queries, and 56p depth maps.

| Metrics | EK100MIR
mAP / nDCG ↑ | EgoMCQ
inter / intra acc ↑ | EK100CLS
top-1 / top-5 acc ↑ | EgoNLQ
mIoU ↑ | EgoMQ
mAP ↑ | DE
scale-aware RMSE / scale-invariant RMSE ↓ |
|---|---|---|---|---|---|---|
| $\mathcal{L}_{vtc}$ | 29.7 / 30.7 | 94.2 / 60.2 | 12.847 / 30.037 | 6.14 | 6.97 | 0.572 / 0.495 |
| $\mathcal{L}_{vtc} + \mathcal{L}_{depth}$ | 31.3 / 31.2 | 94.2 / **62.6** | 15.412 / 32.995 | 5.98 | 7.52 | 0.5637 / **0.464** |
| $\mathcal{L}_{augvtc}$ | 31.3 / 32.2 | 94 / 61.6 | **16.508** / 32.851 | **6.53** | 6.14 | 0.550 / 0.489 |
| $\mathcal{L}_{augvtc} + \mathcal{L}_{depth}$ | **33.1 / 33.1** | **94.6 / 62.6** | 15.898 / **33.895** | 6.17 | **8.87** | **0.539** / 0.481 |

Table 7: Ablation studies on downstream tasks.

## 4.3 Ablation Study

**EgoDTM's components.** Combining video-text matching with depth estimation enhances each learning objective. EgoDTM leverages depth maps to capture object relations, while textual descriptions enriched with shape and movement details can benefit more from depth information. As shown in the table Table 7, progressively adding our modules leads to consistent improvements across the majority of tasks, and our model outperforms all baselines in the main experiments. One potential limitation is that depth pretraining may negatively impact the performance on EgoNLQ to some extent. Nevertheless, our proposed AugVTC module is able to mitigate this effect and yields improvements that surpass those achieved by Avion, LaViLa, and the EgoVLP encoder as shown in Table4 of our main paper. Besides, comparing row 2 and row 4 when adding $\mathcal{L}_{augvtc}$, we observe an increase in the scale-invariant RMSE and a decrease in the scale-aware RMSE. We hypothesize that this may result from the proxy task bias introduced by multi-task pretraining.

**Depth query numbers.** The number of depth queries affects depth granularity. We find that setting the query number to 8 is optimal, with each query covering a moderate depth range within $[0, 1]$.

| Method | mAP
@(0.5:0.95) | map
@0.50 | hand-AP
@(0.5:0.95) | object-AP
@(0.5:0.95) |
|---|---|---|---|---|
| Upperbound [13] | 60.70 | 73.31 | 90.87 | 30.53 |
| Our Pipeline | 43.02 | 54.88 | 61.01 | 25.03 |

Table 8: Evaluation of our HOI mask generation pipeline on HOI segmentation benchmark VISOR [13] in kitchen environment.

**Resolution of the predicted depth map.** The resolution of depth maps greatly impacts GPU memory usage. We find that higher-resolution depth maps slightly improve the video-text alignment, but this comes at the cost of reducing the maximum batch size. Therefore, we choose a resolution to 56p as a balanced trade-off between high resolution and the ability to maintain a large batch size.

**Model size.** A larger model size enhances performance, especially on the EgoMCQ intra task, which requires selecting the correct video from visually similar alternatives.

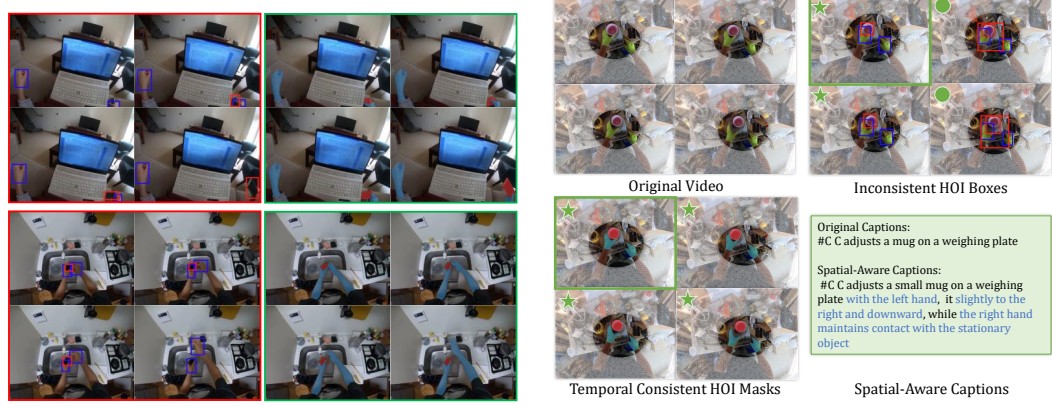

Figure 4: Comparisons of the noisy HOI bounding boxes (left) and the spatial-temporal consistent HOI masks (right).

Figure 5: Example of generalizable data construction. For better visualizations, we blur the background to highlight HOI regions.

**Text sampling strategy.** We examine the impact of text sampling strategy on model performance. A mixed sampling strategy that includes both enriched and original texts for pretraining yields stronger results by leveraging both detailed and simplified textual information.

**Batch size.** The experiment results show that increasing batch size steadily improves performance.

### 4.4 More Analysis

**Quantitative analyses of the HOI mask generation pipeline.** To evaluate the reliability of the HOI detector and SAM2 model in the egocentric domain, particularly for hand-object interactions, we evaluate the image-based HOI segmentation on the VISOR dataset [13]. The results are presented in Table 8. The supervised training model on VISOR serves as the upper bound for performance comparison. While our generation pipeline does not fully reach the upper bound, it demonstrates high-quality segmentation results. Notably, hand segmentation is significantly lower than the upper bound, likely due to inaccuracies in HOI detection prompts and challenges in segmenting hand-object interactions with SAM2 in cluttered egocentric backgrounds.

**Qualitative analyses of the HOI mask generation pipeline.** We firstly compare inconsistent HOI bounding boxes from HOID [63] (frame-by-frame detection) with spatial-temporal HOI masks generated by combining HOID [63] and SAM2 [59] in Figure 4. The HOI boxes detected by HOI detector often lose tracks, since they are detected by an image-based model. Our pipeline achieves consistent HOI tracking across frames, offering more precise HOI labels. Figure 5 presents that our data construction pipeline is useful even when HOI regions are small and the background is noisy.

## 5 Conclusion

In this work, we present EgoDTM, a novel egocentric 3D-aware video-language model. EgoDTM integrates dual transformer encoders with a lightweight 3D-aware depth decoder, trained using video-text contrastive learning and depth estimation objectives. To enable large-scale pretraining, we generate millions of depth maps and spatially enriched captions by leveraging foundation models. The captions are enhanced through a detect-track-generate pipeline specifically tailored for egocentric videos. EgoDTM demonstrates intuitive generalization in estimating depths in unseen environments. Extensive experiments across diverse benchmarks, spanning short video understanding, 3D understanding, and long video understanding, validate the effectiveness of our approach.

**Discussions.** While EgoDTM demonstrates strong performance in egocentric hand-object interaction scenarios, its generalization to broader indoor scenarios remains limited. Further exploration may include integrating 3D-aware visual encoders into multimodal large language models to enhance spatial awareness. Moreover, pretraining large-scale spatial-aware egocentric models with richer 3D signals, as explored in VGGT [71], remains a promising yet challenging direction.

# Acknowledgments

This work was partially supported by the Beijing Natural Science Foundation (No. L233008) and the Outstanding Innovative Talents Cultivation Funded Programs 2024 of Renmin University of China.

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

# A    Implementation Details

## A.1    Background of Foundation Models

**HOI Detector [63].** HOID is a robust system for detecting human hands and their interacting objects in images. It is built upon Faster-RCNN [60], pretrained on 100K image dataset with hand-object interaction annotations, including hand bounding box, interacting object bounding box, hand side (left or right), hand contact state (e.g., no contact, self-contact, other person contact, contact with portable object, or contact with a non-portable object). To enhance its capabilities, HOID is further trained with an additional 42K egocentric data samples, enabling improved understanding of HOI from egocentric view. We leverage HOID to generate spatial HOI boxes, hand side, hand contact state for each video, from 12 uniformly sampled frames per video. However, HOID often produces temporally inconsistent box predictions across adjacent frames. To address this, we apply a robust image and video segmentation model to refine the detection results, ensuring greater consistency and accuracy.

**Segment Anything 2 [59].** SAM2 is a versatile segmentation model capable of segmenting objects in both images and videos according to a given prompt, such as a point, box or mask, with remarkable efficiency. It is trained on a large-scale SA-V dataset, comprising 50.9K videos and 35.5M high-quality masks. SAM2 employs a hierarchical image encoder and a memory mechanism to handle streaming frame input. In our approach, SAM2 is utilized to generate the spatial-temporal consistent HOI masks by leveraging prompts derived from HOID outputs.

**Depth Anything 2 [77].** DAv2 excels in monocular depth estimation, offering fine-grained details, strong generalization and efficient inference. It is built upon the pretrained visual foundation model DINOv2 [46] and a depth decoder DPT [58]. Pretrained on 595K synthetic images and 62M pseudo-labeled real images, DAv2 exhibits strong out-of-domain generalization. In our work, we use DAv2 to generate depth maps for eight frames sampled from egocentric videos, serving as supervision signals. Following the recommendations of DAv2, we employ the DAv2-Large variant, which produces more spatial-temporal consistent depth maps.

**DeepSeek-LLM [14].** We employ the LLM to interpret HOI information and enrich textual descriptions with shape and movement details. DeepSeek-LLM demonstrates exceptional ability to follow instructions and comprehend HOI mask prompts. Specifically, we use the DeepSeek API to access their 200B-parameter LLM to facilitate these tasks.

## A.2    Dataset Details

**Ego4D [22].** Ego4D contains 3,670 hours of egocentric videos with dense narrations, covering diverse scenarios and activities from worldwide. Each narration is timestamped and paired with a free-form sentence. Following the approach in [87], we construct 4M video-text clip pairs for pre-training, with an average clip length of 1 second (±0.9). In our text enrichment process, we only keep those hand-object interaction clips performed by the camera wearer, where the text begins with '#C' (denoting the wearer) and then follows HOI-related verbs and nouns. This strategy excludes clips that record other people's activities, such as multi-person interactions [61] where the text begins with '#O', and the videos like '#C C walks away'. For the natural language query task, it comprises 1,659 untrimmed videos, each averaging 500 seconds in duration. On average, each video contains 12 clip-query pairs. Following the official split from [22], we use 11,291 queries for training and 3,874 for validation.

**Epic-Kitchens [11].** Epic-Kitchens-100 (EK-100) consists of 100 hours of egocentric cooking videos divided into training (67,217 clips), validation (9,668 clips), and testing (13,092 clips) splits. Each clip includes start and end timestamps, a short textual narration, and a verb and noun class that correspond to the narration. There are 3805 action classes, 97 verb classes, and 300 noun classes. We evaluate our pre-trained model on the validation split.

**EGTEA [34].** EGTEA comprises 28 hours of egocentric cooking videos, annotated with 10,321 instances of fine-grained actions across 106 classes. The average action duration is 3.2 seconds. For our experiments, we use only the visual frames as input. We follow prior works [27, 87] and report top-1 accuracy and mean class accuracy on all three test splits, including 2,022 testing instances for each split.

**H2O [29].** H2O is a dataset capturing egocentric hand-object interactions in a laboratory, including 36 action classes. The egocentric data is captured from an Azure Kinect camera mounted egocentrically for recordings. Since our primary target is to evaluate the transfer learning capability of visual representation, the train/val splits have 7862/11638 frames.

## A.3    Experimental Details

**Zero-Shot Video-Text Retrieval in EK100MIR and EgoMCQ.** We perform video-text matching with 16 frames as input for EK100MIR and 4 frames for EgoMCQ, following [86].

**Zero-Shot Action Recognition in EGTEA.** We follow the evaluation protocol proposed by [34] to compute the mean performance across all evaluation splits. This involves performing video-text retrieval between video clips and the action text labels, which are prompted by prepending "#C C ...". During inference, we apply three spatial crops of size $224 \times 224$ from each $256 \times 256$ frame of 10 video clip, averaging predictions across these crops to produce the final results.

**Depth Estimation in H2O.** The frozen visual encoder produces feature maps of dimension 768, which are passed to a linear decoder to estimate depths with a resolution of $720 \times 720$. The model is trained for 10 epochs with a batch size of 64 and a learning rate of 0.0005, where the first 1.5 epochs serve as a warm-up phase. Our evaluation code is built upon Probing3D [18].

**Robot Manipulation in Franka Kitchen.** In Franka Kitchen environment, all baselines apply imitation learning for visuomotor control. A policy network is trained for each task using observations from the environment and video representation from EgoDTM. To adapt EgoDTM and LaViLa, we repeat the image observation 4 times as video input. For each task, the experiment is conducted from two different camera viewpoints for two random seeds using 50 randomly sampled trajectories. The final result is the average of the success rate. Our codebase is built upon MPI [26].

**Natural Language Queries on Ego4D.** The task typically operates on a 6-minute video. Using the video compression technique from [6], we compress the original video frames by 6 times to save storage. Then we extract the features with the fps of 1.87 and sampling frame number 4. We takes 256 dimension global video features and 768 dimension BERT features as input. Our codebase is built upon EgoVLP [36].

# B    Additional Quantitative Results

**Predicted depth contains valuable information for action recognition.** Since ground-truth data is unavailable for directly evaluating the depth decoder, we demonstrate the utility of our predicted depth maps for multimodal action recognition, as shown in **??**. We simply encode the depth map using an MLP at a 56p resolution. The action recognition accuracy improved by +1.4%, confirming that our predicted depths contain meaningful information for unseen egocentric data.

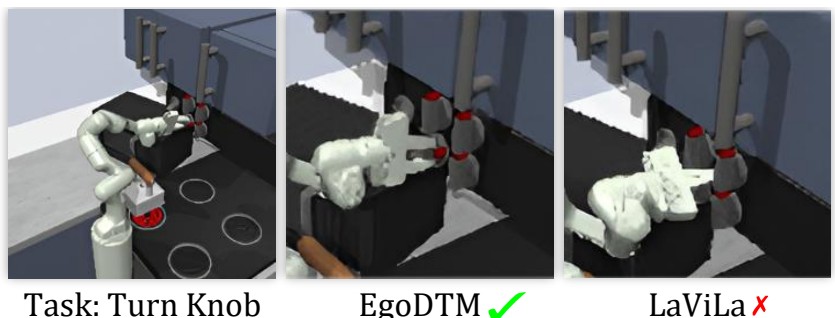

Task: Turn Knob    EgoDTM ✓    LaViLa ✗

Figure 6: Qualitative results of robot manipulations.

## C  Additional Qualitative Results

**LLM Prompts.** We use the LLM prompts in Figure 7 to enrich the texts with HOI shape and movement information. Specifically, the shape information is provided by the HOI mask areas, where the large, medium, small object occupies [0.1,1], [0.01, 0.1] and [0, 0.01] areas, respectively.

**Generated Data.** Examples of generated HOI boxes, HOI masks, depths, and enriched texts are illustrated in Figure 8.

**Case Study on Robot Manipulation.** In Figure 6, the learned policy based on EgoDTM visual representation enables the robot to approach the switch and turn it, while LaViLa successfully approaches but misses the switch.

## D  Discussions

**Comparison with Related Works that Pretrained with Depth.** ImageBind [21] and Language-Bind [88] are the most relevant depth-based vision-language pretraining works. These methods employ multiple encoders to align various modalities within a unified feature space through contrastive learning. While both methods utilize depth for pretraining, their application may be less impactful when applied to conventional third-person datasets. In contrast, depth is essential for egocentric perception, where spatial awareness is critical for understanding human indoor activities. Furthermore, their pretraining processes treat depth as an input rather than a prediction target, resulting in depth features that lack pixel-level 3D information and video representations that remain unaware of 3D structure. In our work, we aim to enable video representations to predict depth maps, thereby embedding 3D awareness directly into the representations.

**Potential for Real-World Applications.** While our model demonstrates improvements over text-pretrained models in both video understanding and robotic manipulation tasks, it falls short of state-of-the-art performance of manipulation models. However, our model predicts more meaningful depth maps in real-world settings than in simulations, offering promising potential for real-world deployment.

## Egocentric Video with HOI Masks

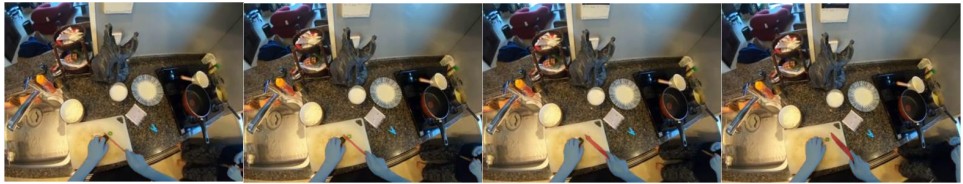

## System Prompt

**## Background**
1. The user will provide information about a short egocentric video (12 frames) captured by one person using VR/AR, approximately one second long, with continuous frame box annotations. The annotations represent the center point and size of objects using the format: <length x, width y, area s, contact state (optional)>, where values range from [0, 1] indicating percentages of the length or width. The area is the product of length and width. The annotations may include up to four elements: the left hand, the right hand, an object related to the left hand, and objects being manipulated by the hands.
2. About the directions, smaller x means more left, larger x means more right, smaller y means more higher, larger y means more lower.
3. If the area is larger than 0.1, then the object is large object; if the area is larger than 0.01 but smaller than 0.1, then the object is medium size; if the area is smaller than 0.01, then the object is small object.
4. There are five possible contact states: no contact, self-contact (between the user's hands), contact with another person, contact with a portable object (e.g., an apple), or contact with a stationary object (e.g., furniture).
5. Note that I can only assure the hands, but the types of left/right can not be guaranteed. Typically, if there exists two hands, the hand with lower x is the left hand, the hand with larger x is the right hand. Another you should notice is that, if there are mostly left hand but exists few right hand data, you should ignore the right hand data, vice versa.

**## Response Requirements**
1. You should interpret the hand-object interaction (HOI) in the video: Use the given text information to describe the interaction process, such as relevant relations, human actions and objects. Describe the hand positions, movement directions, and speed, as well as the sizes and positions of any objects.
2. Remember that the information in original text must be contained in your response.
3. Your response should be a precise, fluent and unified natural language summary, restrictly using less than two sentences. I denote C as the user, please start your response with '#C C ...' .
4. Avoid using pronouns like 'their', 'his', 'her'. Never mention 'in the video', just express what happens.
5. Don't express the same thing twice. Never use parentheses i.e., (), to explain your meaning.
I will provide you the above mentioned information. The infomation will keep empty if the video does not have that type of object.

## Input Prompt

Given the HOI-related information below, respond me with a new sentence following the above requirements.
**Original text**: #C C Cuts a cucumber on a chopping board with a knife.
**Possible left hand**: [].
**Possible right hand**: [[0.59, 0.92, 0.01, 'portable object contact'], [0.96, 0.96, 0.0, 'portable object contact'], [0.98, 0.88, 0.0, 'portable object contact'], [0.59, 0.92, 0.01, 'portable object contact'], [0.96, 0.96, 0.0, 'portable object contact'], [0.98, 0.87, 0.0, 'portable object contact'], [0.59, 0.93, 0.01, 'portable object contact'], [0.97, 0.95, 0.0, 'portable object contact'], [0.98, 0.86, 0.0, 'portable object contact'], [0.59, 0.93, 0.01, 'portable object contact'], [0.97, 0.94, 0.0, 'portable object contact'], [0.98, 0.84, 0.0, 'portable object contact'], [0.58, 0.93, 0.01, 'portable object contact'], [0.98, 0.93, 0.0, 'portable object contact'], [0.98, 0.82, 0.0, 'portable object contact'], [0.58, 0.93, 0.01, 'portable object contact'], [0.98, 0.92, 0.01, 'portable object contact'], [0.98, 0.81, 0.0, 'portable object contact'], [0.58, 0.93, 0.01, 'portable object contact'], [0.98, 0.91, 0.01, 'portable object contact'], [0.98, 0.8, 0.0, 'portable object contact'], [0.58, 0.93, 0.01, 'portable object contact'], [0.98, 0.91, 0.01, 'portable object contact'], [0.98, 0.8, 0.0, 'portable object contact'], [0.58, 0.93, 0.01, 'portable object contact'], [0.98, 0.9, 0.01, 'portable object contact'], [0.98, 0.78, 0.0, 'portable object contact'], [0.59, 0.93, 0.01, 'portable object contact'], [0.98, 0.89, 0.01, 'portable object contact'], [0.98, 0.78, 0.0, 'portable object contact'], [0.59, 0.93, 0.01, 'portable object contact'], [0.98, 0.88, 0.01, 'portable object contact'], [0.98, 0.77, 0.0, 'portable object contact'], [0.59, 0.93, 0.01, 'portable object contact'], [0.99, 0.88, 0.01, 'portable object contact'], [0.97, 0.76, 0.0, 'portable object contact']].
**Hand (not sure which hand)**: [].
**Object1**: [[0.45, 0.77, 0.0], [0.46, 0.78, 0.0], [0.46, 0.78, 0.0], [0.46, 0.79, 0.0], [0.47, 0.79, 0.0], [0.48, 0.8, 0.0], [0.48, 0.8, 0.0], [0.49, 0.8, 0.0], [0.49, 0.8, 0.0], [0.5, 0.8, 0.0], [0.51, 0.79, 0.0]].
**Object2**: [].
**Your response**:

## LLM Response

#C C holds a knife with the right hand, moving it downward to cut a small cucumber on a chopping board.

Figure 7: LLM prompt strategy for generating enriched text from HOI masks and the original text.

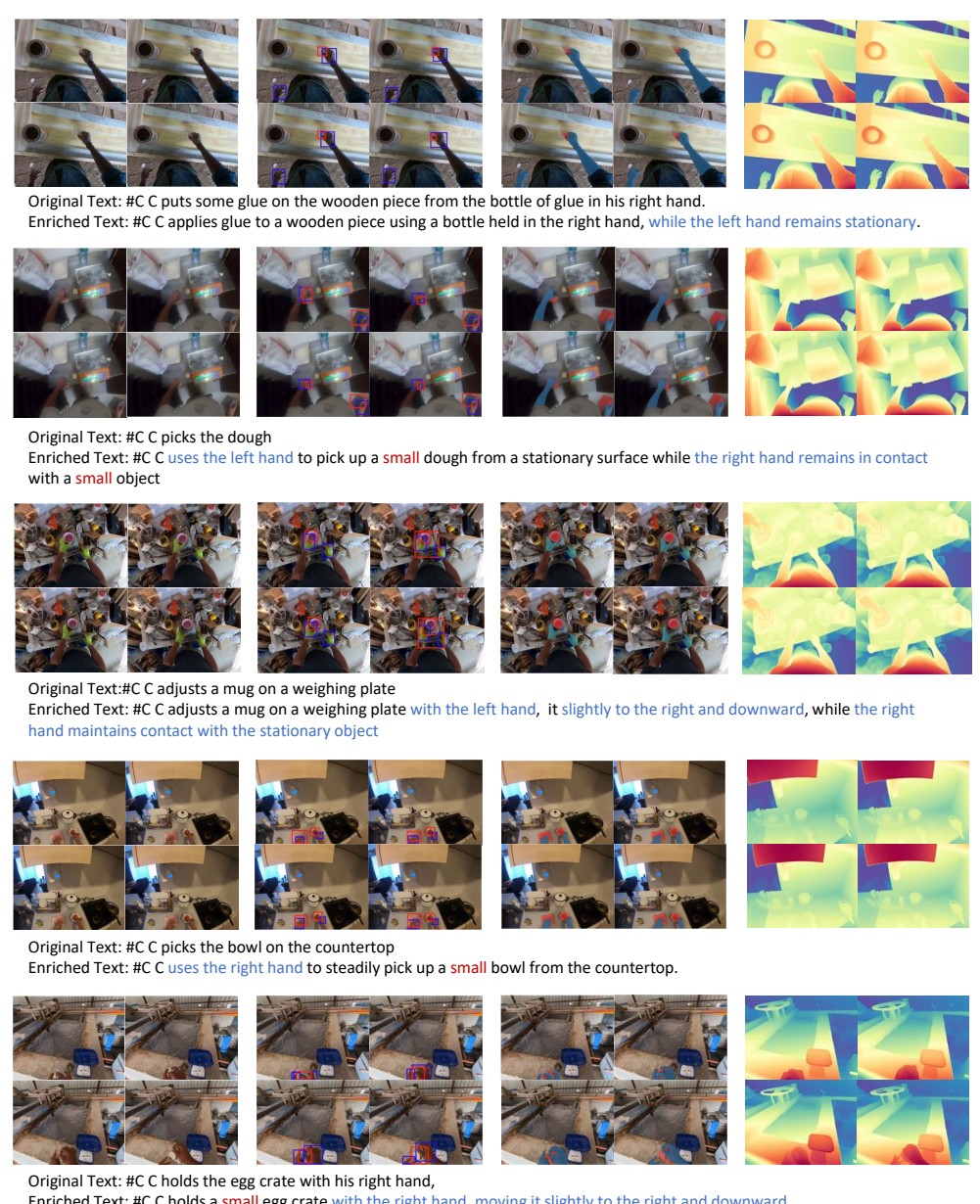

Original Text: #C C puts some glue on the wooden piece from the bottle of glue in his right hand.
Enriched Text: #C C applies glue to a wooden piece using a bottle held in the right hand, while the left hand remains stationary.

Original Text: #C C picks the dough
Enriched Text: #C C uses the left hand to pick up a small dough from a stationary surface while the right hand remains in contact with a small object

Original Text:#C C adjusts a mug on a weighing plate
Enriched Text: #C C adjusts a mug on a weighing plate with the left hand, it slightly to the right and downward, while the right hand maintains contact with the stationary object

Original Text: #C C picks the bowl on the countertop
Enriched Text: #C C uses the right hand to steadily pick up a small bowl from the countertop.

Original Text: #C C holds the egg crate with his right hand,
Enriched Text: #C C holds a small egg crate with the right hand, moving it slightly to the right and downward.

Figure 8: Illustration of data generated by our data generation pipelines, including intermediate HOI boxes, masks, and the enriched texts and depth maps used as supervision signals. The text that includes HOI movements is marked blue, while the contents that include HOI spatial information are marked red.

