# OpenReview forum: "EgoDTM: Towards 3D-Aware Egocentric Video-Language Pretraining"
_NeurIPS.cc/2025/Conference — NeurIPS 2025 poster_

### Official Review · Reviewer_LJgN · 2025-06-27

**Clarity:** 3
**Significance:** 3
**Originality:** 3
**Rating:** 4
**Confidence:** 4

**Summary:**

This paper proposes a novel egocentric video-language pretraining method, which aims to inject 3D-awareness into video-language foundation models. The key design of the proposed EgoDTM is incorporating a lightweight 3D-aware decoder to effciently learn 3D-awareness from pseudo depth maps generated by DepthAnythingV2. Besides, EgoDTM enriches the original short video captions with hand-object visual cues by organically combining several foundation models. Experiments on Epic-Kitchens-100-MIR, EGTEA and EgoMCQ demonstrate the effectiveness of the proposed method.

**Questions:**

- The experiments are conducted on mainly three benchmarks, which is less than previous works like HENASY. HENASY has conducted more experiments on EK100-CLS, EgoNLQ and EgoMQ, thus I think the authors can consider adding more experiments on these benchmarks.
- The proposed method uses DeepSeek-LLM-200B model to enrich video captions. However, DeepSeek-LLM-200B should be an LLM without involving video inputs, how do you generate rich video descriptions? If no video inputs are involved, I think there may be some hallucination.
- The key contribution of the paper is introducing 3D-awareness into egocentric video-language pretraining. However, the paper does not provide experiments specifically designed to show models' 3D-awareness. I suggest the authors to add some experiments, e.g., can the proposed design lead to better understanding of the relationships of objects in terms of front and back.

**Ethical Concerns:**

["NO or VERY MINOR ethics concerns only"]

**Final Justification:**

My concerns have been addressed after rebuttal. Most importantly, the authors provide experiments results on EgoMQ and EK100-CLS to further demonstrate the effectiveness of the proposed model, and they also present a detailed explanation to show that why their experiments can verify the 3D-awareness of their model. I really appreciate their effort in addressing these concerns.

My only concern is similar to that of Reviewer YGLr, i.e., introducing only depth as 3D guidance has limitations. However, I still believe this work represents **a meaningful step forward**.

Overall, I recommend accepting this paper due to its good contributions.

**Limitations:**

yes

**Quality:**

3

**Strengths And Weaknesses:**

Strengths:
- The paper is well written and easy to follow.
- To the best of my knowledge, this paper is the first to introduce 3D-awareness in egocentric video-language pretraining.
- The proposed design based on depth is reasonable, and the experiments demonstrate the effectiveness.

Weaknesses:
- The experiments are limited on mainly three benchmarks, which is less than previous works like HENASY.
- I think the authors should provide some experiments to demonstrate their superiority in understanding 3D scenes, as their main contribution is about 3D-awareness.
- Please refer to the Questions part for more details. If the authors can address my concerns, I am very willing to increase my rating.

---

> ### Author Rebuttal · Authors · 2025-07-31
>
> We greatly appreciate the time and effort you invested in providing these detailed observations and questions. We have carefully considered your comments and outlined our responses and proposed revisions below. We hope these clarifications and revisions address your concerns and further enhance the manuscript.
>
> **W1 & Q1: The paper lacks sufficient downstream task evaluations.**
>
> Thank you for your valuable feedback. First, we sincerely apologize if the formatting caused you to miss our EgoNLQ experiments in Table 4. We also include robot manipulation experiments in Appendix Table 1. Furthermore, we have added new experiments on EgoMQ, as presented below:
>
> |          |        |        |  EgoMQ |        |        |        |         |
> |:--------:|:------:|:------:|:------:|:------:|:------:|:------:|:-------:|
> |          | R1@0.3 | R5@0.3 | R1@0.5 | R5@0.5 | R1@0.7 | R5@0.7 | avg-mAP |
> | AVION35M |  32.17 |  47.3  |  23.11 |  36.3  |  13.45 |  19.6  |   6.23  |
> |  LaViLa  |  32.9  |  48.68 |  **24.12** |  37.59 |  **15.23** |  22.47 |   8.12  |
> |  EgoVLP  |  30.44 |  46.66 |  22.41 |  35.75 |  13.05 |  20.21 |   6.11  |
> |  EgoDTM  |  **32.92** |  **50.08** |  23.94 |  **39.15** |  14.81 |  **22.65** |   **8.87**  |
>
> Besides, we add ablation on action recognition task on Epic-Kitchens-100, i.e., EK100CLS below:
> |          |  EK100-CLS |         |
> |:---------:|:------:|:-------:|
> |          | top-1 acc | top-5 acc|
> |$\mathcal{L}_{\rm{vtc}}$    | 12.84 | 30.03 |
> |$\mathcal{L}\_{\rm{vtc}}+\mathcal{L}_{\rm{depth}}$        | 15.41|32.99  |
> |$\mathcal{L}_{\rm{augvtc}}$    | 16.50 | 32.85 |
> |$\mathcal{L}\_{\rm{augvtc}}+\mathcal{L}_{\rm{depth}}$     | 15.89 | 33.89 |
>
> In our updated version, we will revise the layout to present EgoNLQ and EgoMQ side by side in two columns.
>
> **W2 & Q3：This paper should provide more experiments related to 3D scene understanding.**
>
> This work currently focuses on 3D-awareness centered around human actions from an egocentric view instead of indoor scene understanding, as illustrated in Figure 3. Although our model is capable of providing some depth understanding of surrounding objects / humans (as illustrated in the first Ego4D example in Figure3), it is overall constrained within the scope of egocentric hand-object interactions.
>
> We argue that the benefits of 3D-awareness can be reflected through improved performance on general downstream tasks like short video understanding on EK100MIR, EgoMCQ, EK100CLS [added during rebuttal period], long video understanding on EgoNLQ, EgoMQ [added during rebuttal period], and 3D-aware tasks on H2O, FrankaKitchen [in supplementary]. We plan to further explore more general-purpose 3D-aware visual encoders in future work.
>
> **Q2: How are the captions generated using LLMs? How do you handle hallucinations?**
>
> We carefully prompt an LLM to understand the change of HOI masks to ensure faithfulness compared with original captions. To prove the faithfulness, we evaluate the enriched captions and original captions by calculating the BERTScore[ICLR'20], the results are presented below, which indicates the good quality of our captions.
>
> | Recall             | Precision         | F1                 |
> |--------------------|-------------------|--------------------|
> | 0.818 | 0.546 | 0.652 |
>
> **Regarding the hallucination issue, our perspective is as follows:**
> - First, through careful prompt engineering and manual verification, we found that the LLM can effectively capture changes in bounding boxes and generate corresponding descriptive modifications, while largely preserving the original semantic content.
> - Second, although some hallucinations are present, training with these captions consistently improves performance across most downstream benchmarks, as shown in our ablation studies. This suggests that the benefits outweigh the potential drawbacks.
> - Finally, prior work such as LLaVA [NeurIPS’23] has demonstrated that LLMs are capable of understanding bounding box information and producing fine-grained descriptions, which leads to better generalization. Inspired by this, we believe it is feasible to generate high-quality captions by prompting LLMs to understand HOI mask information. Moreover, we intuitively believe that generating captions using video-based MLLMs (e.g., Gemini-2.5-Pro, QwenVL2.5, InternVL3) is more prone to hallucination compared to our proposed pipeline.

---

> > ### Comment · Reviewer_LJgN · 2025-08-05
> >
> > Thank you for the additional experiments on EgoMQ and EK100-CLS. However, I still have concerns on the 3D-awareness and the LLM-generated captions.
> >
> > 1. Could you please show more details about why the 3D-awareness can be reflected through current experiments? Which parts of these benchmarks need 3D-awareness?
> >
> > 2. Could you please show more details about "prompt an LLM to understand the change of HOI masks to ensure faithfulness"? In addition, I think the BERTScore cannot be used to evaluate long captions. Please clarify these.

---

> > > ### Author Response · Authors · 2025-08-05
> > > **Response to Reviewer LJgN's Comments**
> > >
> > > Thank you for your thoughtful responses. We hope to address your concerns and further enhance the manuscript as follows:
> > >
> > > **Q1: Why is 3D awareness (also referred to as depth awareness) essential for current benchmarks?**
> > >
> > > First, we explain intuitively why 3D awareness is essential for egocentric perception and then follows the explanation for relevant tasks.
> > >
> > > **Argument 1: Egocentric perception inherently requires 3D spatial perception, either implicitly learned as representation like EgoDTM or explicitly use depth as inputs for models.**
> > >
> > > - **Intuitive importance of 3D awareness under indoor environment** (line 35-37 in our paper): Egocentric tasks typically involve scenarios where a person wearing an AR/VR headset or a camera interacts within indoor environments, such as cooking in a kitchen or doing laundry. In these scenarios, models equipped with depth perception can learn meaningful indoor priors specific to the environment. This 3D awareness provides rich spatial cues, enabling the model to inherently understand and reason about the layout and geometry of the indoor space.
> > >
> > > - **Motivation of introducing 3D awareness to CLIP (line 37-41 in our paper)**: If the model only perceives traditional RGB text-aware features like CLIP or EgoVLP, it is expected that these models fail to identify the transition of adjacent events in the same environment, while 3D-aware features can alleviate this issues. Prior works such as EyesWideShut[CVPR'24] have demonstrated that CLIP features lack such 3D awareness. Therefore we aim to prove that injecting direct 3D supervision like depth could brings benefit toward visual encoders.
> > >
> > > **Argument 2: The effectiveness over other egocentric encoders on current benchmarks reflect the effectiveness of 3D-aware video pretraining.**
> > >
> > > Before explaning specific tasks, consider the following scenario: a person is preparing lunch in a kitchen. The VR video input often contains complex background, with the HOI region occupying only a small portion of the pixels. As the person transitions between actions, e.g., from cutting onions to picking up tomatoes, the camera moves, causing view changes in the background, while the true structure of the environment remains consistent.
> > >
> > > - **For short videos understanding containing EK100MIR, EgoMCQ and EK100CLS**, 3D depth-awareness enables the model to disambiguate fine-grained EgoHOI from complex backgrounds, recognizing ``cutting onions'' by implicitly emphasizing HOI regions (line 45-46 in our paper).
> > >
> > > - **For long video understanding including EgoNLQ and EgoMQ**, 3D-awareness helps detect specific actions from long 3D-aware video features, such as detecting ``cutting onions'' video clips from 3D-aware long video features.
> > > All depth information of the seen environment could be obtained from these video features, indicating that 3D structures lie in the long video features.
> > >
> > > - **For spatial-aware tasks like depth estimation on H2O, and robot manipulation task on FrankaKitchen**, accurate reasoning about the distance between image content and the camera is essential in such cases.
> > >
> > > **Lastly, we emphasize that 3D awareness is a broad concept. Through this work, we aim to provide an initial attempt and empirical validation for incorporating 3D-awareness into egocentric video pretraining, and we hope it lays the foundation for further exploration in this direction.**
> > >
> > > **Q2: Details of the LLM prompting process, and the evaluation of captions.**
> > >
> > > We have included the detailed process in **Supplementary Figure 3 on Page 5**. And more cases are shown in **Supplementary Figure 4 on Page 6**. As shown, the captions are enriched from the original ones of “#C C cuts a cucumber on a chopping board with a knife,” to a more fine-grained one: “#C C holds a knife with the right hand, moving it downward to cut a small cucumber on a chopping board.” To our best effort, our system prompt helps the model produce relatively faithful output without losing the original meanings.
> > >
> > > **Regarding the BERTScore evaluation beween enriched captions and original captions**:
> > > As seen in our provided common cases in Supplementary, the captions are often not too long. We agree that the optimal way for evaluation might be the GPT-based evaluators. But the BERTScore shows a high recall of 0.818, which suggests that the enriched captions capture a substantial portion of the key semantics present in the original captions.

---

> ### Comment · Reviewer_LJgN · 2025-08-06
>
> Thanks for the authors' detailed explanation. Most of my concerns have been addressed. I decide to raise my rating to borderline accept.
>
> In addtion, I strongly encourage the authors to include these details in the paper, especially the 3D awareness of current benchmarks. In my understanding, the 3D awareness is directly reflected by the spatial-aware tasks like depth estimation on H2O, not by videos understanding tasks. This is because these video benchmarks could be improved due to other factors, e.g., better understanding of objects and contexts. Furthermore, including other details can improve the quality of the paper.

---

> > ### Author Response · Authors · 2025-08-06
> > **Response to Reviewer LJgN's Comments**
> >
> > Dear reviewer LJgN,
> >
> > Thank you for acknowledging our efforts in addressing your concerns. Your constructive suggestions have been instrumental in improving the quality of our work. We sincerely appreciate the revised score, thank you for recognizing our improvements.
> >
> > Following you suggestions, we will incorporate all relevant details during rebuttal, such as more downstream experiments and explanations on 3D awareness, to our main paper.
> > We will also open-source all our code, model weights, and the associated data generation pipelines.
> >
> > Best regards,
> >
> > Authors

---

### Official Review · Reviewer_pB7n · 2025-06-29

**Clarity:** 3
**Significance:** 2
**Originality:** 2
**Rating:** 4
**Confidence:** 3

**Summary:**

To enhance the understanding of egocentric scenarios, the authors propose EgoDTM, a video-language model equipped with 3D-aware capabilities. The model incorporates a lightweight 3D-aware decoder, which learns spatial structure information from pseudo-depth maps generated by a depth estimation algorithm, under the supervision of pseudo-depth labels provided by a teacher model. Additionally, EgoDTM integrates several foundational modules—such as hand detection and object recognition—to enrich the original concise textual descriptions with semantic cues of hand-object interactions. The effectiveness of EgoDTM is demonstrated across various tasks, including video-text retrieval, action recognition, depth estimation, robotic manipulation, and long video understanding.

**Questions:**

1) Lack of in-depth ablation studies on the 3D-aware video pretraining architecture. The current evaluation only includes an ablation on 𝐿_depth , which is insufficient to thoroughly and clearly demonstrate the advantages and underlying mechanism of the proposed 3D-aware decoder.

2) Concerns about the impact of the teacher model on performance. Since the enhancement relies on incorporating DepthAnything as a teacher model, it remains unclear how much of the performance gain is attributable to this component. This raises the question of whether a simpler knowledge distillation approach could achieve similar improvements in 3D perception without introducing complex structures.

3) More detailed ablation on the depth estimation task of the H2O dataset would provide stronger evidence. As this task directly reflects the model's 3D perception capability, conducting thorough ablation on key components (e.g., enhanced captions, 3D-aware decoder) on the H2O dataset would offer clearer validation of each module’s contribution to 3D understanding.

**Ethical Concerns:**

["NO or VERY MINOR ethics concerns only"]

**Final Justification:**

After reading the authors' rebuttal, the authors have addressed my major concerns. I will raise my score.

**Limitations:**

The limitation mentioned by the authors is solely that the method has not yet been explored in broader domains.

**Paper Formatting Concerns:**

No problem.

**Quality:**

2

**Strengths And Weaknesses:**

Strengths:
1) Clear motivation and sound methodology. The authors aim to bridge the modality gap in video-language pretraining by introducing depth supervision, incorporating HOI detection, keyframe selection, and bidirectional tracking, and generating spatially-grounded captions using large language models. This motivation is well-justified and practically relevant.

2) Well-structured and clearly written. The paper is logically organized, and the methods and experiments are presented in a complete and comprehensible manner.

Weaknesses:
1) Limited ablation study. The paper only conducts a coarse ablation on 𝐿_vtc, 𝐿_depth, and 𝐿_augvtc. While the difference between 𝐿_vtc and 𝐿_augvtc clearly demonstrates the benefit of spatially-enriched text, the actual impact of the 3D-aware decoder is unclear. Its effect is only reflected by whether 𝐿_depth is applied or not, which fails to provide insightful evidence for the value and effectiveness of this module.

2) Potential fairness issue with the teacher model. To generate pseudo-depth labels, the authors incorporate DepthAnything as a teacher model during 3D-aware video pretraining. This could lead to unfair comparisons with other methods. Moreover, since the role of the 3D-aware decoder is not sufficiently validated, it raises the question of whether the gains are primarily due to the teacher model itself. For instance, replacing the current pretraining strategy with a simpler knowledge distillation approach might yield similar results—this possibility is not ruled out.

3) Lack of direct evaluation of 3D perception enhancement. Both the enhanced textual descriptions and the proposed 3D-aware decoder are designed to improve 3D perception. However, no ablation is conducted on the depth estimation task of the H2O dataset, which would have been crucial to explicitly identify which components most effectively contribute to 3D understanding.

---

> ### Author Rebuttal · Authors · 2025-07-31
>
> We greatly appreciate the time and effort you invested in providing these detailed observations and questions. We have carefully considered your comments and outlined our responses and proposed revisions below. We hope these clarifications and revisions address your concerns and further enhance the manuscript.
>
> **W1 & Q1：The role of the 3D-aware decoder is unclear.**
>
> $L\_{\rm{depth}}$ refers to the combination of the 3D-aware decoder and the associated depth loss ($L\_{\rm{depth}}$); both components are either jointly added or removed. Therefore, the ablation of $L\_{\rm{depth}}$ illustrates the effectiveness of the 3D-aware video pretraining process.
>
> During rebuttal, we sincerely apologize that we might not be able to conduct more pretraining experiments, but In the revised version, we promise to include additional ablations by replacing the 3D-aware decoder with alternatives such as a lightweight MLP decoder or a ConvNet decoder to further validate the effectiveness of our 3D-aware decoder.
>
> Nonetheless, In this work, our primary contribution is the introduction of 3D-awareness into EgoVLP, demonstrating its general applicability across a wide range of tasks and achieving performance that surpasses previous methods.
>
> **W2 & Q2: Potential Unfair Comparison of Using Depth Anything as Teacher Model**
>
> We first provide a comparison of different methods' pretraining process as follows:
>
> |              | EgoVLP | EgoVLPv2 | AVION | LaViLa | HelpingHands |  HENASY  |   EgoDTM   |
> |:------------:|:------:|:--------:|:-----:|:------:|:------------:|:--------:|:----------:|
> |  Supervision |  Text  |   Text   |  Text |  Text  |   Text+Box   | Text+Box | Text+Depth |
> |   Text Num   |   4M   |    4M    |  35M  |   35M  |      35M     |    35M   |     6M     |
> |    Params    |  172M  |   363M   |  169M |  142M  |     206M     |   277M   |    151M    |
> | EK100MIR-mAP |  23.3  |   30.8   |  32.9 |  30.8  |     31.2     |   31.3   |    33.5    |
>
> - **First, all methods are evaluated fairly during inference.** We ensure fair comparisons by using the same original video-text pairs and maintaining comparable model sizes. However, it is both common and inevitable for different approaches to incorporate distinct foundation models for pseudo-label generation during pretraining. For example, LaViLa utilizes T5-Large for text rewriting, while HelpingHands employs an HOI detector to obtain HOI bounding boxes.
> - Moreover, **different methods aim to prove the effectiveness of different types of supervision signals**, as illustrated in Figure 1 of our paper, often relying on powerful teacher models tailored to their objectives. For instance, LaViLa and AVION leverage a multimodal large language model and LLM to synthesize ten times more text data, i.e., 35M text data for supervision, while we use only 6M text data in total for pretraining. HelpingHands[ICCV'23] and HENASY[NeurIPS'24] not only generates HOI boxes using an HOI detector but also adopts HOI detection as an auxiliary pretraining task.
> - **Finally, our objective is not to distill knowledge from a specific foundation model, but rather to learn from noisy 3D supervision signals directly**. As such, knowledge distillation is not the focus of this work. Depth supervision derived from any depth estimation foundation model, DepthPro [ICLR'25], can serve the same purpose. But we admit that there exists bias from the DepthAnything model. In future work, we plan to include additional experiments using depth maps generated by DepthPro to further validate our approach.
>
> **W3 & Q3: Lack of ablation studies for depth estimation on H2O.**
>
> Thank you very much for your suggestion. Following your advice, we have conducted ablation studies on almost all downstream tasks including H2O. The results are as follows:
>
> |                                              | EK100MIR     | EgoMCQ          | EK100CLS         | EgoNLQ | EgoMQ |Depth                                  |
> |----------------------------------------------|---------------|-----------------|------------------|--------|-------|---------------------------------------|
> | metrics                                      | mAP / nDCG↑   | inter / intra acc ↑| top-1 / top-5 acc ↑| mIoU   ↑| mAP   ↑|scale-aware RMSE / scale-invariant RMSE ↓ |
> | $\mathcal{L}_{\rm{vtc}}$                          | 29.7 / 30.7  | 94.2 / 60.2     | 12.847 / 30.037  | 6.14   | 6.97  |0.572 / 0.495                          |
> | $\mathcal{L}\_{\rm{vtc}}+\mathcal{L}_{\rm{depth}}$      | *31.3* / 31.2| *94.2* / **62.6**| 15.412 / *32.995* | 5.98   | *7.52* |*0.5637* / **0.464**                   |
> | $\mathcal{L}_{\rm{augvtc}}$                       | *31.3* / *32.2*| 94 / 61.6       | **16.508** / 32.851| **6.53**| 6.14  |0.550 / 0.489                          |
> | $\mathcal{L}\_{\rm{augvtc}}+\mathcal{L}_{\rm{depth}}$   |  **33.1** / **33.1** | **94.6** / **62.6** | *15.898* / **33.895** | *6.17* | **8.87** |**0.539** / *0.481*                    |
>
> As shown, incorporating our components consistently improves the scale-aware RMSE metric, indicating that our method enhances the understanding of the true depth. Notably, when comparing row 2 and row 4, where enriched text is added, we observe an increase in the scale-invariant RMSE, which indicates impairing the understanding of relative depth structures.
> Generally, the two components proposed in this work generally lead to improvements on the majority of 2D vision tasks, indirectly demonstrating the effectiveness of 3D-awareness.

---

### Official Review · Reviewer_YGLr · 2025-06-30

**Clarity:** 3
**Significance:** 3
**Originality:** 2
**Rating:** 4
**Confidence:** 3

**Summary:**

The paper proposes an egocentric video pretraining model that enhances traditional video-language models with 3D awareness. The pretraining process introduces a lightweight 3D-aware decoder, supervised by incorporating 3D depth information. Additionally, the authors enrich traditional textual descriptions with spatial awareness. Experimental results demonstrate that the proposed pretraining approach yields improved performance across several downstream tasks, including depth estimation, language-based video queries, video-text retrieval, and action recognition.

**Questions:**

1. How much does it cost to generate the enhanced dataset using all these foundation models? How does the author envision generalizing this process to web-scale video data or general video pretraining?

2. Given the limited improvement in the numerical results, can the author provide an in-depth analysis of common failure cases where the 3D-aware video features still fail to resolve the issues?

**Ethical Concerns:**

["NO or VERY MINOR ethics concerns only"]

**Final Justification:**

Thank you for the response. All of my concerns have been addressed with the additional results. In the rebuttal, the authors have shown promising potential to extend the training data generation pipeline at an acceptable cost, and they plan to include more failure case analyses as suggested. The proposed method, EgoDTM, makes solid contributions by enhancing video pretraining with both enriched captions and depth information. While using only depth has limitations in fully representing the 3D world—including aspects like object location and orientation—the method still represents a meaningful step forward. I will keep my positive rating.

**Limitations:**

Yes.

**Paper Formatting Concerns:**

No.

**Quality:**

3

**Strengths And Weaknesses:**

**Strengths**

The paper addresses a significant limitation in egocentric video understanding by incorporating 3D depth information. The proposed method effectively integrates off-the-shelf depth estimation models for supervision and enriches video captions with spatially-aware information through a detect-track-generate pipeline.

**Weakness**

One concern is that while the improvements across benchmarks are consistent, they are often modest (typically around 1–2%). Another concern lies in the resource requirements for generating the pretraining data. The method heavily depends on multiple foundation models (e.g., DepthAnything, SAM2, HOI-D, and large language models), which may limit scalability for broader or more general video pretraining applications.

---

> ### Author Rebuttal · Authors · 2025-07-31
>
> We greatly appreciate your thoughtful assessment and comments on our paper. Below, we address the concerns in detail:
>
> **W1: The performance improvements across benchmarks seem marginal.**
>
> We argue that our improvements are non-trivial. As shown in the ablation study in Table 3(a), EgoDTM achieves a +3.4 mAP gain over the vanilla baseline on the EK-100-MIR benchmark, improving from 29.7 to 33.1. Furthermore, on the depth estimation task, our model achieves a 9.8% improvement in Scale-aware RMSE over the best-performing baseline (LaViLa), as shown in Table 2. We enumerate the additional supervision signals employed by the baselines and present comparisons to demonstrate that the improvements achieved by our method are substantial rather than modest.
>
> **Q1: How long does it take to generate the training data? And how do you consider the scalability of the approach?**
> - At inference time, depth map generation takes approximately 0.45 seconds per 4-frame 720p video, using 4 GB of GPU memory. Generating HOI bounding boxes and masks takes around 4.96 seconds per 4-frame 720p input, with a memory footprint of 3.8 GB. Therefore, generating 2.5 million training samples requires approximately 159 A100 GPU hours for HOI bounding box and mask generation, and 15 A100 GPU hours for depth map generation. For LLM text generation, we spent $418 USD and approximately 8 hours of API usage to generate 2.5 million captions.
>
> |           Evaluated on 1 A100 card           |    Depth    | HOI Box+HOI Mask |
> |:--------------------------------------------:|:-----------:|:----------------:|
> |          Time per 4-frame video (s)          |    0.45s    |       4.96s      |
> |                  GPU Mem (G)                 |      4G     |       3.8G       |
> |    Total A100 GPU Hours for 2M data (hour)   |     15h     |       159h       |
>
> - From a scalability perspective, we believe this is a highly cost-effective and easily extensible paradigm. The cost of pseudo-label generation is relatively low, and our 3D-aware decoder is lightweight. We have already demonstrated the feasibility of this approach on the million-sample scale, and scaling it to tens of millions is within reach.
>
> **Q2: Common failure cases of EgoDTM.**
>
> Based on our observations, EgoDTM achieves notable improvements over the base model in action recognition and is better able to attend to HOI-relevant regions in complex scenes. Nonetheless, it still suffers from a relatively high error rate in challenging scenarios, such as those involving extreme viewpoints or occlusions of human-object interactions. We plan to include the corresponding qualitative examples in the appendix of the revised version.

---

> > ### Comment · Reviewer_YGLr · 2025-08-05
> >
> > Thank you for the detailed explanation, especially regarding the cost of data preparation. I would recommend that the authors include these details in the paper (either in the main text or the supplementary material), as data generation is a key factor for reproducibility.

---

> > > ### Author Response · Authors · 2025-08-06
> > > **Response to Reviewer YGLr's Comments**
> > >
> > > Dear reviewer YGLr,
> > >
> > > Thank you for your suggestions. In the revised version, we will integrate the content from the rebuttal into the main text and further polish the entire paper. We will also open-source all our code, model weights, and the associated data generation pipelines.
> > >
> > > Best regards,
> > >
> > > Authors

---

> ### Comment · Reviewer_YGLr · 2025-08-06
>
> Thanks for the effort! I will keep my positive rating.

---

### Official Review · Reviewer_kLWJ · 2025-07-02

**Clarity:** 4
**Significance:** 3
**Originality:** 3
**Rating:** 4
**Confidence:** 3

**Summary:**

The research paper introduces EgoDTM, an Egocentric Depth- and Text-aware Model designed to address the lack of 3D understanding in existing video-language pretraining frameworks. Current models often rely on 1D text or 2D visual cues, but humans perceive a 3D world. To bridge this gap, EgoDTM jointly trains a model using both video-text contrastive learning and a novel 3D-aware video pretraining task. This task incorporates a lightweight 3D-aware decoder that efficiently learns to predict depth from pseudo-depth maps generated by powerful depth estimation models. To further enhance spatial reasoning, the system enriches brief, original captions with detailed hand-object interaction (HOI) information through a "detect-track-generate" pipeline that leverages a combination of foundation models. Extensive experiments show that EgoDTM achieves superior performance across diverse downstream tasks, including video-text retrieval, action recognition, and robot manipulation, highlighting its improved 3D-aware visual understanding.

**Questions:**

The questions are listed in the weaknesses section. If the author could solve the concerns, I would consider raising my score. Besides, I want to know whether the paper is not written in the full 9 pages.

**Ethical Concerns:**

["NO or VERY MINOR ethics concerns only"]

**Final Justification:**

Depth estimation as proxy task may not be very strong technical contribution. But the result of the robot manipulation experiments is a strong support. I maintain my original score.

**Limitations:**

The model and its entire pretraining data pipeline are highly specialized for egocentric video, with a strong emphasis on hand-object interactions (HOIs). While this leads to state-of-the-art performance in this domain, it raises concerns about the generalizability of the learned representations to third-person video or other video domains that do not feature prominent hands and close-up object manipulation. The model may be overfitting to the specific biases of the egocentric perspective, potentially limiting its utility as a general-purpose video representation model. The authors acknowledge this as a limitation, but its significance as a trade-off for specialization warrants emphasis.

**Quality:**

3

**Strengths And Weaknesses:**

Strengths:
1.The paper addresses a well-motivated and significant limitation in video-language understanding—the absence of 3D spatial reasoning.
The core idea of integrating a depth-prediction task as an auxiliary objective for pretraining is effective, as demonstrated by strong empirical results.

2.The design of the lightweight 3D-aware decoder is a key strength. It resolves the tension between the high-resolution requirements of dense prediction tasks (like depth estimation) and the large-batch requirements of contrastive learning.

3.The empirical evaluation is thorough and convincing. The authors test their model on a wide array of downstream tasks, including zero-shot retrieval, action recognition, depth estimation, and temporal localization. Consistent and significant performance gains over strong baselines across all these tasks strongly support the paper's claims.

4.The paper is well-written, clearly structured, and easy to follow. Figure 2, in particular, provides an excellent and comprehensive illustration of the entire pretraining framework.

Weakness
1. The paper defines 3D awareness as the ability to infer depth from its representations. While practical, this is a narrow definition of 3D understanding. The model is trained to regress a 2.5D depth map provided by a teacher model. This raises the question of whether EgoDTM is learning fundamental geometric principles or simply becoming very good at mimicking the output of the teacher model (DepthAnythingv2). True 3D reasoning involves understanding object permanence, volume, relative pose, and interaction affordances, which are not directly measured or enforced by the current depth-prediction task. The improvements on downstream tasks could stem from better foreground-background segmentation (a side-effect of depth prediction) rather than a deeper, more holistic 3D scene understanding.

2. The enriched captions generated by the pipeline are not only more spatially aware but also significantly longer and more descriptive than the original captions. Could the performance improvement in retrieval tasks be partially attributed to this increased textual detail, rather than purely to the improved spatial accuracy? An analysis that attempts to disentangle the effects of caption length/detail from spatial correctness would strengthen this contribution.

3. The paper suggests that its contributions are beneficial for downstream tasks like "robot manipulation" in line 82. However, the experiments do not include any tasks on a physical robot or in a simulated robotic environment. There remains a significant sim-to-real (or in this case, perception-to-action) gap, and therefore, the claim of improving robot manipulation is speculative and not directly obtained by the provided results.

4. The overall setting utilized in the ablation study should be further clarified for easier performance comparison. Besides, it would be better if the author could conduct the ablation study on other tasks (not merely the short video understanding ( video-text retrieval and action recognition).

---

> ### Author Rebuttal · Authors · 2025-07-31
>
> We greatly appreciate the time and effort you invested in providing these detailed observations and questions. We have carefully considered your comments and outlined our responses and proposed revisions below. We hope these clarifications and revisions address your concerns and further enhance the manuscript.
>
> **W1：The proxy task of depth estimation may not lead the model to truly understand 3D information. And downstream tasks do not include such true 3D tasks.**
>
> -  **On the choice of proxy task**: We acknowledge that depth estimation may not fully capture all aspects of true 3D understanding. Richer 3D supervision (such as from multi-view geometry or dynamic 3D correspondences in videos) could provide stronger inductive signals. However, our work represents an initial attempt to incorporate 3D supervision into large-scale egocentric video-language pretraining. Introducing 3D signals and training procedures—whether through data collection or model design—is highly non-trivial at the million-scale pretraining level. Despite these challenges, our results show that depth estimation serves as a practical and effective proxy, enabling the encoder to become more 3D-aware and motivating further research in this direction.
> - **On downstream task evaluation**: We agree that evaluating a visual encoder's physical 3D reasoning capability is essential. However, we argue that there are currently few suitable benchmarks to directly evaluate the encoder's capability. Instead, we choose to demonstrate that 3D-aware encoders have potential for solving 2D perception tasks like short video understanding on EK100MIR, EgoMCQ, **EK100CLS [added during rebuttal period, see response to W3 below]**, long video understanding on EgoNLQ, **EgoMQ [action detection task, added during rebuttal period, see table below]**, and 3D-aware tasks on H2O, FrankaKitchen [in supplementary].
>
> |          |        |        |  EgoMQ |        |        |        |         |
> |:--------:|:------:|:------:|:------:|:------:|:------:|:------:|:-------:|
> |          | R1@0.3 | R5@0.3 | R1@0.5 | R5@0.5 | R1@0.7 | R5@0.7 | avg-mAP |
> |  EgoVLP  |  30.44 |  46.66 |  22.41 |  35.75 |  13.05 |  20.21 |   6.11  |
> |  LaViLa  |  32.9  |  48.68 |  **24.12** |  37.59 |  **15.23** |  22.47 |   8.12  |
> | AVION |  32.17 |  47.3  |  23.11 |  36.3  |  13.45 |  19.6  |   6.23  |
> |  EgoDTM  |  **32.92** | **50.08** |  23.94 |  **39.15** |  14.81 | **22.65** |   **8.87**  |
>
>
> **W2：Is the performance gain in the downstream retrieval task mainly attributed to longer captions?.**
>
> No.
> - First, for downstream vision-language tasks such as retrieval, **we only utilize the encoder during inference**. As shown in Figure 2, no caption augmentation is performed at inference time. Therefore, the improved retrieval performance reflects the model’s enhanced video-text alignment, which is acquired during training with our enriched captions.
> - Furthermore, we highlight that **both baseline methods, LaViLa and AVION, leverage LLMs and MLLMs to expand and rewrite the original captions**, resulting in 35 million enriched descriptive captions. In contrast, we only have 2.5 million newly generated captions informed by HOI masks. Despite using significantly fewer enriched captions, our method outperforms both baselines.
> - To follow your advice, we promise to conduct a pretraining ablation as follows: generate enriched longer and descriptive captions using LLMs without spatial information, to eliminate the impact of spatial-awareness in texts. Due to space and resource constraints, we have not included such pretraining ablation in this version, but we promise to add them to our revised version.
>
> **W3: Do you have experimental support for robot manipulation?**
>
> Yes, we apologize for the confusion. In Appendix Table 1, we present results from robot manipulation experiments conducted in the Franka environment. EgoDTM outperforms baselines including CLIP, ConvNext and LaViLa, demonstrating the potential value of EgoDTM encoder in robotic applications.Some beneficial results are presented below:
>
> |        |   TK   |  OD |   OM   |   FS   |   SD   | Average |
> |--------|:------:|:---:|:------:|:------:|:------:|:-------:|
> | ResNet |   28%  | 18% | 26.70% |   50%  | 75.50% |  39.70% |
> |  CLIP  | 26.30% | 13% | 24.70% | 41.70% | 86.30% |  38.40% |
> | LaViLa |   48%  | 26% | 22.50% |   69%  | **94.50%** |   52%   |
> | EgoDTM |   **56%**  | **28%** | **35.50%** |   **81%**  | 92.50% |  **58.60%** |
>
> **W4: Could you conduct ablation studies on more tasks?**
>
> Yes. We have conducted additional ablation studies on almost all tasks to further validate the effectiveness of our model. During rebuttal, we add experiments including action recognition task on Epic-Kitchens-100 (EK-100-CLS) and action detection task on Ego4D (EgoMQ). The results are presented below:
>
> |                                              | EK100MIR     | EgoMCQ          | EK100CLS         | EgoNLQ | EgoMQ |Depth                                  |
> |----------------------------------------------|---------------|-----------------|------------------|--------|-------|---------------------------------------|
> | metrics                                      | mAP / nDCG↑   | inter / intra acc ↑| top-1 / top-5 acc ↑| mIoU   ↑| mAP   ↑|scale-aware RMSE / scale-invariant RMSE ↓ |
> | $\mathcal{L}_{\rm{vtc}}$                          | 29.7 / 30.7  | 94.2 / 60.2     | 12.847 / 30.037  | 6.14   | 6.97  |0.572 / 0.495                          |
> | $\mathcal{L}\_{\rm{vtc}}+\mathcal{L}_{\rm{depth}}$      | *31.3* / 31.2| *94.2* / **62.6**| 15.412 / *32.995* | 5.98   | *7.52* |*0.5637* / **0.464**                   |
> | $\mathcal{L}_{\rm{augvtc}}$                       | *31.3* / *32.2*| 94 / 61.6       | **16.508** / 32.851| **6.53**| 6.14  |0.550 / 0.489                          |
> | $\mathcal{L}\_{\rm{augvtc}}+\mathcal{L}_{\rm{depth}}$   |  **33.1** / **33.1** | **94.6** / **62.6** | *15.898* / **33.895** | *6.17* | **8.87** |**0.539** / *0.481*                    |
>
> As shown in the table, progressively adding our modules leads to consistent improvements across the majority of tasks, and our model outperforms all baselines in the main experiments. One potential limitation is that depth pretraining may negatively impact the performance on EgoNLQ to some extent. Nevertheless, our proposed AugVTC module is able to mitigate this effect and yields improvements that surpass those achieved by Avion, LaViLa, and the EgoVLP encoder as shown in Table4 of our main paper. Besides, comparing row 2 and row 4 when adding $\mathcal{L}\_{\rm{augvtc}}$, we observe an increase in the scale-invariant RMSE and a decrease in the scale-aware RMSE. We hypothesize that this may result from the proxy task bias introduced by multi-task pretraining.
>
> **Q: The paper is not written in the full 9 pages.**
>
> Thanks for pointing this out. In the current submission, we aimed to balance content between the main paper and the supplementary material and some detailed results and analyses were moved to the supplementary to avoid exceeding the page limit. However, this led to underutilization of the main paper space. In the revised version, we will reorganize the content to make full use of the 9-page limit, including adding the comprehensive ablation study across downstream tasks as well as results on the EK-100-CLS and EgoMQ benchmark.

---

> > ### Comment · Reviewer_kLWJ · 2025-08-05
> >
> > Thank you for the additional information. After considering the response, I will keep my original score.

---

> > > ### Author Response · Authors · 2025-08-06
> > > **Responses to Reviewer kLWJ's Comments**
> > >
> > > Dear reviewer kLWJ,
> > >
> > > We appreciate your constructive suggestions, which have been instrumental in enhancing the quality of our work. In the revised version, we will incorporate the content from the rebuttal into the main text and further refine the entire manuscript. We will also open-source all our code, model weights, and the associated data generation pipelines.
> > >
> > > Best regards,
> > >
> > > Authors

---

### Note · Authors · 2025-08-14

Dear AC and Reviewers,

We sincerely appreciate your valuable time and insightful feedback.
We are encouraged by the reviewer's recognition of our clear motivation (**kLWJ**, **pB7n**), significance of the addressed problem (**kLWJ**, **YGLr**), novel paradigm (**LJgN**), effective and sound method design (**kLWJ**, **YGLr**, **pB7n**, **LJgN**). thorough and strong evaluation(**kLWJ**), well-structured writing (**kLWJ**, **pB7n**, **LJgN**)

During the rebuttal phase, we have addressed the raised concerns and further strengthened our work with additional experiments. We also greatly appreciate that three reviewers have given positive ratings.

In summary, our rebuttal focuses on the following key points:
- **Clarification of the key contribution of 3D-aware 2D model (kLWJ, pB7n,  LJgN, YGLr)**: Humans are capable of implicitly perceiving 3D information from 2D visual inputs. We emphasize that 3D awareness in 2D models is still a broad concept without a unified definition. In this work,we demonstrate that learning 3D depth-awareness for 2D visual features not only improves performance on 2D egocentric tasks but also yields gains on certain 3D tasks. Across multiple downstream benchmarks, our method surpasses strong baselines such as EgoVLP, LaViLa, AVION, which is a non-trivial achievement.
- **Additional validations on more benchmarks (kLWJ,  LJgN)**: We further evaluate our approach on action recognition (EK100CLS) and action detection (EgoMQ). We also benchmark the performance of the current EgoVLM encoder for comparison. The results show that EgoDTM remains the best-performing model.
- **More fine-grained ablations (kLWJ, pB7n)**: We conduct ablation studies on EK100CLS, EgoMQ, EgoMCQ, EgoNLQ, and DepthEstim, analyzing in detail the contributions of our two key designs, including spatial-aware text enrichment and 3D-aware video pretraining, across all downstream tasks. The results confirm that both designs consistently yield notable performance gains on most tasks.

We will integrate all these experiments and analyses into the final version, further clarifying our design choices, highlighting our technical contributions, and strengthening our overall argument.

We believe our work introduces an innovative, effective, and broadly validated video-language paradigm for egocentric vision. We will open-source all code, datasets, and benchmarks to foster progress in the community.

Best regards,

Authors

---

### Decision · Program_Chairs · 2025-09-17

**Decision:**

Accept (poster)

**Comment:**

In their initial ratings, the reviewers were split: two leaned toward rejection and two toward acceptance.

Reviewers agreed that the paper addresses an important limitation of egovideo understanding models, that is the lack of 3D awareness. The authors propose a reasonable pretraining method that uses an off-the-shelf depth estimator to supervise a lightweight 3D encoder. The submission is well written, and the experiments across multiple benchmarks show the effectiveness of the approach. However, reviewers raised several concerns, including: The need for ablations to separate the effects of generated captions versus spatial awareness; missing evidence of improved performance in robot manipulation tasks; potential scalability issues; missing evaluation of improved 3D perception.

In the rebuttal, the authors provided new experiments and results that effectively addressed these concerns. After rebuttal, all reviewers are leaning toward acceptance, with reviewers LJgN and YGLr noting that while relying only on 3D awareness has limitations, the work still represents a meaningful step forward.

The AC agrees with the reviewers consensus and recommends acceptance. The AC also asks the authors to include the new ablations and results from the rebuttal in the final version for completeness.